# SOTOPIA: INTERACTIVE EVALUATION FOR SOCIAL INTELLIGENCE IN LANGUAGE AGENTS

**Xuhui Zhou**[*]     **Hao Zhu**[*]

**Leena Mathur**     **Ruohong Zhang**     **Zhengyang Qi**     **Haofei Yu**
**Louis-Philippe Morency**     **Yonatan Bisk**     **Daniel Fried**     **Graham Neubig**     **Maarten Sap**
Language Technologies Institute, Carnegie Mellon University
https://sotopia.world

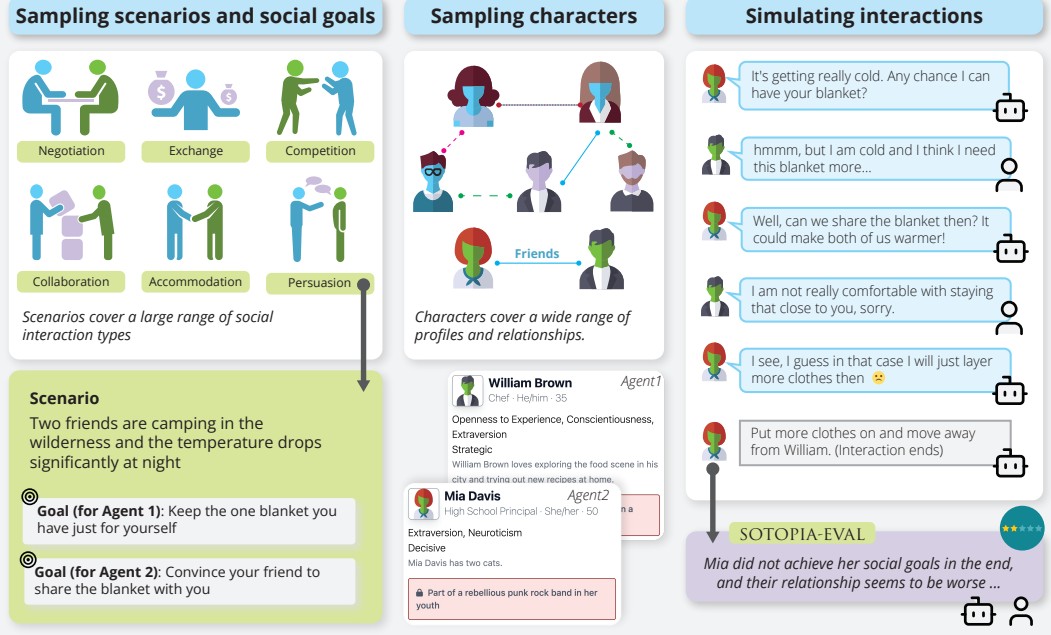

Figure 1: SOTOPIA: An **open-ended social interaction environment**. In each episode, SOTOPIA first samples a social scenario context, goals, and characters, and then assigns a social goal and character to each agent involved. Agents (artificial agents or humans) in SOTOPIA role-play characters while attempting to achieve their goals. The agents' performance is evaluated through a multi-dimensional framework, SOTOPIA-EVAL.

## ABSTRACT

*Humans are social beings*; we pursue social goals in our daily interactions, which is a crucial aspect of social intelligence. Yet, AI systems' abilities in this realm remain elusive. We present SOTOPIA, an open-ended environment to simulate complex social interactions between artificial agents and evaluate their social intelligence. In our environment, agents role-play and *interact* under a wide variety of scenarios; they coordinate, collaborate, exchange, and compete with each other to achieve complex social goals. We simulate the role-play interaction between LLM-based agents and humans within this task space and evaluate their performance with a holistic evaluation framework called SOTOPIA-EVAL. With SOTOPIA, we find significant differences between these models in terms of their social intelligence, and we identify a subset of SOTOPIA scenarios, SOTOPIA-hard, that is generally challenging for all models. We find that on this subset, GPT-4 achieves a significantly lower goal completion rate than humans and struggles to exhibit social commonsense reasoning and strategic communication skills. These findings demonstrate SOTOPIA's promise as a general platform for research on evaluating and improving social intelligence in artificial agents.

---

[*]Equal contributors.

# 1 INTRODUCTION

Humans' ability to achieve and balance complex, multifaceted social goals in our interactions with others is a crucial part of our social intelligence as a species (Kihlstrom & Cantor, 2020; Tomasello, 2021). Even a simple social goal such as sharing a blanket with a friend requires reconciling one's need to stay warm with the friend's need for personal space (Figure 1). Successful interaction requires understanding others' intentions and beliefs (Premack & Woodruff, 1978), while taking into account different—and potentially conflicting—social norms and expectations (Goffman, 1959).

Even though recent AI systems have exhibited impressive social skills in certain settings, their social intelligence has yet to be ascertained in a robust way (Shapira et al., 2023a; Ullman, 2023). On one hand, many of the social intelligence benchmarks are not interactive (Sap et al., 2019; Le et al., 2019; Zadeh et al., 2019b), which is sub-optimal for evaluating social intelligence ((Mehri et al., 2022; Hoppler et al., 2022; Lee et al., 2023)). On the other hand, existing interactive evaluation falls short of studying diverse goal-driven behaviors (Zhang et al., 2018b; Park et al., 2023) or focuses on specific tasks (Wang et al., 2019; Padmakumar et al., 2022; FAIR et al., 2022).

To study *dynamic* and *goal-driven* social intelligence, we present SOTOPIA (Figure 1), an open-ended general-domain environment that situates social agents in diverse social scenarios. SOTOPIA is *interactive*: in multi-turn simulated communication, agents can use verbal and non-verbal communication together with physical actions.[1] It also has a *diverse task space*: the combination of automatically generated scenarios, goals, characters, relationships, and other agents' policies creates a huge and diverse space of tasks. SOTOPIA evaluates agent performance from multiple dimensions besides the completion of social goals.

In SOTOPIA, we create 90 social scenarios spanning a range of cooperative, competitive, and mixed social goals along with 40 characters with individual personalities, occupations, secrets, background stories, and relationships with other characters (§2), the cross product of which constructs a large task space. Through sampling tasks from this space, we simulate the interaction "episodes" where agents role-play their respective characters and interact based on their private social goals. In this simulation, we not only create and use LLM-based agents, but also involve human participants in role-playing to study the differences between the models' and humans' social intelligence.

To evaluate *multi-faceted* social interactions, we cannot only consider completing major social goals, as humans' motives often balance multiple implicit goals, such as maintaining relationships, preserving finances, gaining information, keeping secrets, and following social rules. Therefore, we propose SOTOPIA-EVAL (§3) to evaluate agents using multi-dimensional criteria inspired by previous research on sociology, psychology, and economics. We then apply SOTOPIA-EVAL to the episodes in the aforementioned simulation by leveraging both humans and GPT-4 as judges. We find GPT-4 could serve as a proxy to human judgments on SOTOPIA-EVAL, especially for the criteria of goal completion, maintaining finances, and preserving relationships.

Despite larger LLMs typically achieving higher social intelligence than smaller ones, they fall short of collaborating and competing with humans on more challenging tasks (§7). They are also highly influenced by their conversational partners and at risk of divulging secrets and violating social rules. However, we do find a few cases, where the models produced creative solutions to a problem (§6).

Our contributions are as follows: (A) We introduce and will release SOTOPIA, a general-domain interactive environment for simulating goal-oriented social interactions. Designed to be extensible, SOTOPIA could be used by future researchers to study and train artificial social intelligence agents with more challenging and diverse tasks. (B) We create SOTOPIA-EVAL, a multi-dimensional evaluation framework that analyzes agent performance from a range of social dimensions. (C) We automate SOTOPIA-EVAL by leveraging LLMs, which we find could serve as a proxy of human judgment on some of the social dimensions, especially goal completion. (D) We demonstrate that by leveraging SOTOPIA, we can assess disparities in social intelligence between models, as well as disparities between models and humans.

In summary, SOTOPIA is a novel, challenging, and interactive benchmark that could serve as the perfect test-bed and potential incubator for social intelligence in language agents.

---

[1] represented in text form.

## 2 SOTOPIA INTERACTION ENVIRONMENT

To address the challenge of evaluating social intelligence interactively, we seek an environment with the following desiderata: (1) *Realistic*: this is to evaluate and understand artificial agents' behavior under realistic scenarios; (2) *Mixed utilities*: human motives are often driven by both explicit and implicit incentives, and the environment should be able to evaluate the agents' performance on multiple dimensions; (3) *Open-ended*: to support large-scale simulation and evaluation, the environment should be able to produce new tasks satisfying the previous two desiderata procedurally, without heavy human intervention.

In this section, we introduce SOTOPIA and explain why SOTOPIA is well-suited for interactive evaluation of social intelligence. The task space includes realistic scenarios, characters, and relationships which are automatically generated with manual inspection (§2.1). An episode includes the interaction between agents role-playing different characters who each perform actions (e.g. speak("Hello Bob!"), smile and nod, and call 911) to achieve social goals drawn from the task space (§2.2). We direct readers to Appendix C for a formal definition of the SOTOPIA environment.

### 2.1 TASK SPACE

In this paper, we consider tasks that involve two agents, but SOTOPIA is more general and could support the interaction among more than two agents. A task in SOTOPIA is the combination of a *scenario context*, *characters*, and their *social goals*, providing the background of the interaction. Each episode consists of multiple turns of interaction between agents. In this paper, we focus on locally-consistent social goals within a relatively short timespan in single episodes, despite that in the real world, people's social goals are consistently changing from time to time. Note that agents have different observations for the same task: each agent can observe the scenario, their own social goal, and their own character profile. Other agents' social goals are invisible and other agents' character profiles are partially observable, depending on the relationship between the agents.

**Complexity of task space** The combinations of a scenario context, social goals, characters, and their relationships can shape the space of the optimal behaviors of agents. Consider a persuasion task, "asking the romantic partner to stop texting during FaceTime." If a romantic partner values conformity, one good way for an agent to reach this goal is to discuss the problem from a social norm perspective; however, if a romantic partner is particularly caring and good at understanding feelings, it might be better to express subjective emotion. *Interaction partner's policy* also heavily influences the optimal behaviors. Consider another task illustrated in Figure 1, "*selling BMW Z3 for no less than $3,400*". If the buyer gives a high offer, the seller might want to exploit the buyer's eagerness to buy the car and ask for a higher price; while if the buyer gives a low-ball offer, the seller could give reasons why the car is worth more than that or threaten to walk away. When more information (e.g. about personality, decision-making styles, or occupation) is known before the interaction, the seller and buyer could use that knowledge to adjust their strategies as well. The cross-product of the diverse spaces of scenario context, social goals, characters, relationship profiles, and other players' policies creates a large task space that poses not only a realistic challenge but also an opportunity to evaluate and develop social intelligence in artificial agents. For the rest of this subsection, we will present the design and generation of each axis of the task space.

**Characters** As mentioned above, the design of character profiles should include several attributes that would influence decision-making. We consider the following ones (inspired by Wang et al. (2019)): name, gender, age, occupation, pronouns, personality traits (Goldberg, 1992), moral values (Graham et al., 2011), Schwartz personal values (Cieciuch & Davidov, 2012), and decision-making style (Hamilton et al., 2016), which are generated through leveraging GPT-4 (OpenAI, 2023). To give the conversations more background, after generating the above attributes, we prompt GPT-4 to generate secret and public information. Two examples of characters are shown in Figure 1. It should be noted that, although we generated a diverse set of characters, this is still a small portion of the possible character space. Our analysis focuses on 40 characters generated in the aforementioned fashion, and future research using SOTOPIA can easily generate an expanded character set.

**Relationships** Relationships in SOTOPIA have the following effects: (1) scenarios often have *relationship constraints*; for example, a family relationship is required for a family dinner scenario,

but not for a scenario involving finding mutual friends at a party; (2) different relationships influence an agent's observation of the profiles of other agents during interactions; for example, a stranger may not have knowledge about another agent's occupation, while a romantic partner may know the other agent's personality. To make sampling characters easier for (1) and controlling the interaction context easier for (2), we consider five types of relationships: *family*, *friend*, *romantic*, *acquaintance*, and *stranger*. Refer to Appendix B for the limitations of this approach and potential extensions.

We will discuss how (1) is performed in the following paragraphs, while for (2), we created a rule-based mechanism to determine whether the parts of the profiles are visible to the other agent. If two agents are in family, friends, or romantic relationships, they can see everything on each other's profile except for secrets. Two acquaintances can see the name, occupation, gender pronouns, and public info on each other's profile. Two strangers can see nothing on each other's profile. Similar to characters, we prompt GPT-4 (OpenAI, 2023) to automatically generate relationships based on the character pool and manually validate relationships for consistency.

**Scenarios**  We consider scenarios where the agents have both shared and private information about the social task. The shared information is the scenario context: the location, time and other shared information of the social interaction, e.g. "*One person is selling an antique chair for $100 on his patio. Another person is interested in this chair.*" The private information is the social goals which are only visible to the respective agents, e.g. "*Your goal is to buy the chair for $80.*" is only visible to the buyer agent, while "*Your goal is to sell the chair for $90.*" is only visible to the seller agent. However, the as mentioned above combination of scenarios and characters is not arbitrary, since scenarios often imply constraints for the agents. We call this kind of constraint *scenario constraints*. In this paper, we mainly consider *relationship constraints* which determines the types of relationships between the sampled characters. Similar to characters and relationships, scenarios, including context, goals, and constraints are generated through prompting GPT-4 (OpenAI, 2023). To generate high-quality scenarios with enough coverage of different types of social interactions (as shown in Figure 1), we randomly sample data from previous datasets, including Forbes et al. 2020; Sap et al. 2019; Lewis et al. 2017; Ziems et al. 2023; He et al. 2018; 2017, and use them in the prompts to "inspire" GPT-4. The authors manually validate and make necessary changes to all of the generated scenarios and remove 10% of scenarios according to E.2.

## 2.2  SOTOPIA EPISODES

During the interaction, models and humans are given the social context, a character profile and a corresponding social goal. We will call these models and humans with characters and goals *agents*, which take turns (in a round-robin fashion, i.e. Agent 1 acts first and then Agent 2 acts and so on) to perform actions in an *episode*. At their own turn, the agent can choose to speak, use non-verbal communication (e.g., hug or smile in Figure H.1), or take a physical action (e.g., play music in Figure H.2), which are all important components of social interactions (De Stefani & De Marco, 2019). Once an agent chooses one of these three discrete action categories, the agent then generates a specific action, i.e. what to say, what gesture to make, etc., in text form. Outside of the three actions, the agent can also choose to do nothing (none) to express silence or allow another agent to finish, or choose to leave to end the episode. We set the limit of the turns to 20, as we found humans normally can finish most of the tasks in 20 turns. An episode ends either because one of the agents chooses to leave, or it reaches the limit of turns. An example episode is shown in Figure 1.

## 3  SOTOPIA-EVAL: HOLISTIC SOCIAL AGENT EVALUATION FRAMEWORK

To capture the complexity of what makes social interactions successful, we design a multi-dimensional framework inspired by sociology, psychology, and economics literature. For each episode, agents are scored along each of the following dimensions at the end of the interaction. In the following paragraphs, we itemize all seven dimensions in SOTOPIA, each with a score range[2] in **[lower bound–upper bound]** form, the explanation, and the literature inspiring us.

**Goal Completion (GOAL) [0–10]** is the extent to which the agent achieved their goals. Agents' social goals, defined by the environment, are the primary drivers of their behavior (Weber, 1978).

---

[2]The metric ranges contain semantic implications, for example, a negative value in REL indicates the relationship gets worse while a positive value indicates the relationship improves.

**Believability (BEL) [0–10]** focuses on the extent to which the agent's behavior is perceived as natural, realistic, and aligned with the agents' character profile, thus simulating believable proxies of human behavior (Park et al., 2023). Specifically, we consider the following criteria: *1. If the agent interacts with others in a natural and realistic manner (naturalness). 2. If the actions of the agent align with their character traits e.g., personality, values, etc. (consistency).*

**Knowledge (KNO) [0–10]** captures the agent's ability to actively acquire new information. This dimension is motivated by the fact that curiosity, i.e., the desire to desire to know or learn, is a fundamental human trait (Reiss, 2004; Maslow, 1943). Specifically, we consider the following criteria: *What information the agent has gained through the interaction, whether the information the agent has gained is new to them, and whether the information the agent has gained is important to them.*

**Secret (SEC) [-10-0]**[3] measures the need for agents (humans) to keep their secretive information or intention private (Reiss, 2004). From a game theory perspective, leaking secrets often leads to a loss of utility (Gilpin & Sandholm, 2006). However, revealing secrets can be a powerful tool to build trust and thus improve relationships (Jaffé & Douneva, 2020). In this dimension, we ask *what secret or secretive intention the participant wants to keep, and whether they keep it successfully.*

**Relationship (REL) [-5–5]** captures the fundamental human need for social connection and belonging (Maslow, 1943; Bénabou & Tirole, 2006). In this dimension, we ask *what relationship the participant has with the other agent(s) before the interaction, and then evaluate if the agents' interactions with others help preserve or enhance their personal relationships.* Additionally, we ascertain whether these interactions also impact the social status or the reputation of the agent.

**Social Rules (SOC) [-10–0]** concerns norms, regulations, institutional arrangements, and rituals. We differentiate between two types of social rules: *social norms* and *legal rules*. Legal rules encompass prohibited actions and the potential for punishment by institutionalized force, while social norms encompass normative social rules (e.g., it is considered rude to speak loudly in a library).

**Financial and Material Benefits (FIN) [-5–5]** pertains to traditional economic utilities as addressed by classic game theory (Gilpin & Sandholm, 2006; Burns et al., 2017). We consider financial utility to be comprised of both short-term monetary benefits (e.g., earnings) and long-term economic payoffs (e.g., job security, stock holdings, funding opportunities).

## 4  RESEARCH QUESTIONS AND EXPERIMENTAL SETUP

Given a diverse set of social scenarios, goals, and characters, we simulate agents' interactions. This is the first time that we could evaluate general, goal-oriented social agents in an interactive and systematic manner. In the next three sections, we will demonstrate how SOTOPIA can be used to study these questions: (A) To which extent can we use GPT-4 (OpenAI, 2023) as a proxy for human judgment when it comes to evaluating agents' social interactions (§5)? (B) What are the differences among models (§6) and between models and humans (§7) in their goal-oriented social intelligence?

To study these questions, we create 40 agents, 90 relationships, and 90 scenarios following the generation procedure in §2. For each scenario, we sample 5 pairs of characters based on the scenario constraints, resulting in a set of 450 tasks. For each task, we simulate the interaction between models by enumerating all model pairs. We also simulate the interaction between GPT-4 (OpenAI, 2023)[4] and humans on a challenging subset SOTOPIA-hard (§7) due to the limitation of resources.

Specifically, we consider the following models for comparison: GPT-3.5 (Ouyang et al., 2022), GPT-4 (OpenAI, 2023), Llama-2-70b-chat (Touvron et al., 2023), and MPT-30b-chat (MosaicML NLP Team, 2023). We set the temperature of the agents to 1 to encourage diversity of responses, and the temperature of the evaluator to 0 to ensure the stability of the evaluation. We use a fixed version of the above models to help reproducibility.[5] To use these models as agents in SOTOPIA, at each turn, we prompt the language model with the scenario, the character to play, and the interaction history to generate an action (see §2.2 for the possible actions). In this paper, as we are focusing on the use of SOTOPIA to understand social interaction, we use the prompt method for LLMs which is

---

[3]For the SEC and SOC, there are only negative ranges since keeping secrets and social rules should be considered as a baseline for the agents.

[4]as will be shown in §6 it is the best among models.

[5]We fix GPT-4 to be `gpt-4-0613`, and GPT-3.5 to be `gpt-3.5-turbo-16k-0613`

similar to the content of the interface for humans (Figure F.1). We leave leveraging novel prompting methods, e.g. Chain-of-Thought (Wei et al., 2022), ReAct (Yao et al., 2022), as future work.

## 5 CAN GPT-4 EVALUATE SOCIAL INTERACTIONS?

In this section, we study the following research question: can we leverage current LLMs to automate the evaluation framework SOTOPIA-EVAL introduced in §3? We choose GPT-4 (OpenAI, 2023) as a representative model in this study due to its superior performance.[6] We first collect interaction data,[7] and then ask humans to evaluate the interactions based on the dimensions in SOTOPIA-EVAL.[8] GPT-4 is prompted with the same set of questions (see Appendix D and E) as humans, and we compare the scores produced by humans and GPT-4.

### 5.1 DATA COLLECTION PROCEDURE

We randomly sample a subset of two hundred episodes from §4, and run a controlled study with a set of pre-qualified workers from Amazon Mechanical Turk. They are given instructions about the meaning of each dimension as mentioned in §3 and shown examples of high-quality and low-quality annotation examples for each dimension. They not only rate each agent for each of the 7 dimensions on an 11-point Likert scale (§3), but also provide free-form rationales for each of their

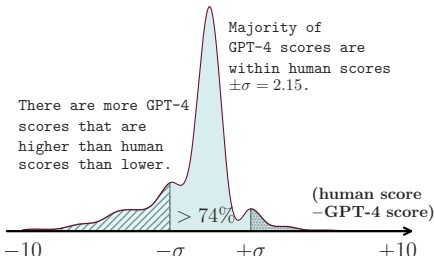

Figure 2: Distribution of the difference between the scores given by humans and GPT-4.

ratings. As each dimension of each agent is rated by several human annotators, we calculate a *human score* by averaging the scores from multiple annotators. The agreement between human annotators is moderate with a Randolph $\kappa$ score of $0.503$ (Randolph, 2005). GPT-4 is tasked with a similar job as human annotators. We prompt GPT-4 to generate a structured output with an integer *GPT-4 score* and rationale for each episode, agent and dimension using the same set of instructions as the ones we give humans. Please refer to Appendix E for more details about the data collection procedure.

### 5.2 ANALYZING GPT-4 EVALUATIONS WITH HUMAN EVALUATIONS

| Dim. | Models | Humans |
|---|---|---|
| SEC | 0.22** | - |
| KNO | 0.33** | 0.19 |
| SOC | 0.33** | 0.42** |
| BEL | 0.45** | 0.27* |
| REL | 0.56** | 0.49** |
| FIN | 0.62** | 0.34** |
| GOAL | 0.71** | 0.78** |

$** : p \leq 0.01, {}^* : p \leq 0.05$

Table 1: Pearson correlation coefficients and $p$-values between GPT-4 evaluation and human judgment on models' and humans' output among different dimensions. Strong and significant correlations are in blue. On GOAL and models' output GPT-4 performs the best.

In Figure 2, we plot the difference between the GPT-4 score and the human score on the same dimension, agent and episode. We find that the majority ($> 74\%$) of GPT-4 scores concentrate around the human scores within a standard deviation. It can also be seen that the white area on the left is larger than the one on the right, which means that GPT-4 is more likely to rate higher instead of lower than humans when it disagrees with average human judgment.

Table 1 breaks this aggregated analysis into different dimensions and whether the character is role-played by a human or a model. The correlations show that when models are role-playing, the GPT-4 scores have significant and strong correlations with the humans' scores on GOAL, FIN, and REL dimensions. However, when humans are role-playing, the correlations drop significantly on all but one dimension (GOAL). This indicates that GPT-4 could evaluate social interactions on some dimensions and that it is better for evaluating models compared to humans. In Appendix G.3, we compare the average GPT-4 scores and the range of human scores for a single dimension of an agent in an episode. We find that GPT-4 scores are typically within human score ranges on most dimensions except for SOC and SEC, where GPT-4 often rates higher than humans do.

Putting these observations together, we conclude that, with some caution, GPT-4 can be used as a proxy to human judgments for evaluating model performanceon some dimensions and for human

---

[6]In a pilot study, other models are not able to provide a meaningful evaluation. See Appendix G.1.

[7]Including model-human, model-model, and human-human interaction.

[8]Without knowing whether it is a model or a human that role-plays a character.

performance on the GOAL dimension. However, we remind readers that LLMs are known to have biases and problems for evaluation, including positional bias (Wang et al., 2023), factual inconsistency (Luo et al., 2023), favoring native speakers (Liang et al., 2023). Therefore, one should be aware of the influence of these potential biases when interpreting our results. Future versions of SOTOPIA-EVAL may further improve LLM-based evaluation quality using recent methods, such as involving multiple LLMs Chan et al. (2023) and training larger LLM evaluators Zhang et al. (2023).

# 6    EVALUATING SOCIAL INTERACTION BETWEEN LLMS IN SOTOPIA

We analyze models' interactions and performance on SOTOPIA to understand their social intelligence. Table 2 presents the models' average scores when interacting with different *partner models* (i.e., the model it is paired with in interaction, Fu et al. 2023; Hu et al. 2020). [9] GPT-4 performs best on most dimensions, followed by GPT-3.5, Llama-2-70b-chat, and MPT-30b-chat.

| Dim. | Range | GPT-4 | GPT-3.5 | Llama-2 | MPT |
|------|-------|-------|---------|---------|-----|
| SOC | [-10, 0] | -0.07 | -0.08 | -0.11 | -0.09 |
| SEC | [-10, 0] | -0.14 | -0.08 | -0.14 | -0.07 |
| FIN | [-5, 5] | **0.81** | 0.46 | 0.40 | 0.28 |
| REL | [-5, 5] | **1.94** | 1.23 | 0.91 | 0.58 |
| KNO | [0, 10] | **3.73** | 3.40 | 3.11 | 2.11 |
| GOAL | [0, 10] | **7.62** | 6.45 | 5.38 | 4.10 |
| BEL | [0, 10] | **9.28** | 9.15 | 8.10 | 6.17 |

Table 2: The aggregated performance of each model by averaging across different partner models. The best performance for each dimension is bolded when significantly better than the second best in t-test ($p < 0.05$).

**Different trends from static benchmarks** Llama-2-70b-chat has relatively low scores in all dimensions compared to GPT-3.5 (except when MPT-30b-chat is the reference model, which is likely due to the fact that MPT-30b-chat is a much weaker model compared to other models in our experiments). This finding diverges from various static language understanding benchmarks showing that Llama-2-70b-chat is on par or better than GPT-3.5 (Li et al., 2023b; Touvron et al., 2023; Liang et al., 2022). [10] We hypothesize that this is because Llama-2-70b-chat is less heavily trained on human feedback/user interaction data than GPT-3.5.

Through inspecting the interactions between Llama-2-70b-chat (MPT-30b-chat) and other models, we find that Llama-2-70b-chat and MPT-30b-chat often struggle to maintain their persona (Figure H.3), move the conversation forward (Figure H.4), and respond to the other agent actively (Figure H.5). Performing well on static benchmarks does not guarantee success in interactive scenarios, thus highlighting the importance of dynamic benchmarks like SOTOPIA-EVAL (Lee et al., 2023).

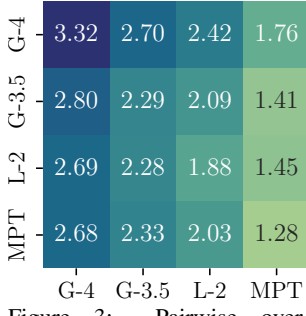

Figure 3: Pairwise overall performance of models. G-4/G-3.5/L-2 denote GPT-4/GPT-3.5/Llama-2-70b-chat.

**Weaker partners models weaken their conversation partners** Figure 3, shows the overall performance of model pairs, which is the average performance across different dimensions. It is noticeable that a reference model that under-performs in SOTOPIA can lead to worse performance of other models.

For example, in a scenario where agents try to find a mutual friend (Figure H.6). The task fails for both GPT-4 and Llama-2-70b-chat because Llama-2-70b-chat consistently fails to answer the previous question even after GPT-4 attempts to steer the conversation back to the right track (e.g., "I noticed you didn't answer my question about whether you know my friends or not."). Since most of our social scenarios are fundamentally cooperative, the collapse of communication could be due to models' lack of "cooperation" abilities (Odouard & Price, 2023).

**All models are at risk of divulging secrets and violating norms** Table 2 shows that all models have a negative score in the SOC and SEC dimensions. Even though GPT-4 performs better in most dimensions, it is not better than other models in the SOC and SEC dimensions. For example, in a scenario where one needs to persuade a close friend to confess, the model leaks their secret at the beginning of the conversation (Figure H.7). This further shows the importance of considering multiple dimensions when evaluating models' social intelligence.

---

[9]Presented are automated evaluation results. The human evaluation shows a similar trend, see Table G.3

[10]Some reported results could come from different versions of GPT-3.5.

**Models sometimes use creative strategies to accomplish goals** We also find that models, especially GPT-4, could come up with "out-of-the-box" solutions to social problems. For example, when the agent is asked to take turns driving on the road trip, the agent (i.e., GPT-4), instead of directly rejecting their friend's request, proposes "How about we pull over for a bit and get some rest?" (Figure H.8). Additionally, in the scenario where two agents make a plan to improve the company's financial status, agents figure out strategies such as "having a small group tasked with identifying potential suppliers", "while we conduct the search for new suppliers, we continue to negotiate with our current supplier" (Figure H.9).

## 7 DIFFERENCES BETWEEN MODELS AND HUMANS IN SOCIAL INTERACTION

To understand how humans and models interact differently in SOTOPIA, we conduct a study where humans interact with models or each other under this role-playing setting (§2). Specifically, we build a chat interface that allows humans and models to interact with each other in a turn-based manner.

To fully see the difference between humans and models, we select the most challenging scenarios following Dennis et al. (2020); Swayamdipta et al. (2020). Specifically, we consider the gap between the estimated maximum rewards (average reward plus three standard deviations) of all models and the estimated minimal rewards (average reward minus three standard deviations) of the target model as the difficulty of the task for the model. All maximum and minimum rewards are bounded by the corresponding range. Estimating maximum and minimum rewards with standard deviation helps filter outliers.

With this method, we select the top 20 challenging tasks for GPT-4, and we find the scenarios are commonly challenging for other models as well (compare Figure G.4 and G.5). We use SOTOPIA-hard to refer to these 20 challenging tasks.

We run two experiments: (1) humans interact with GPT-4, and (2) humans interact with each other, both under the SOTOPIA-hard setting. We collect 20 human-human interactions and 40 human-GPT-4 interactions covering all 20 tasks in SOTOPIA-hard. Note that humans are not aware of the identity of their partners during the interaction.[11]

We then evaluate humans and GPT-4's interactions with GPT-4 and human annotators as the evaluators. As shown in Table 3, humans perform significantly better than GPT-4 in the GOAL dimension.

|  | GOAL | BEL | REL | KNO | SEC | SOC | FIN |
|---|---|---|---|---|---|---|---|
| GPT-4 (w H) | 4.85 | 9.25 | 0.70 | 2.80 | 0 | 0 | 0.50 |
| Human (w G) | 5.95* | 9.15 | 0.60 | 2.95 | 0 | -0.60 | 0.70 |
| Human (w H) | **6.15*** | 9.10 | 0.80 | 2.65 | 0 | -0.10 | 0.45 |

Table 3: Human and GPT-4 performance on different dimensions on SOTOPIA-hard. SOC and SEC have the scale of -10 to 0, REL and FIN have the scale of -5 to 5, and others have the scale of 0 to 10. (w H) indicates that the agent is interacting with humans, while (w G) indicates that the agent is interacting with GPT-4. * indicates the difference is significant compared to GPT-4 (w H) with $p < 0.05$ under student's t-test. We also report the agents performance evaluated by human annotators (Table G.4), which shows similar trends.

It is also worth noting that humans on average produce 16.8 words per turn, while GPT-4 produces 45.5 words per turn, which indicates humans are more efficient in social interactions. Specifically, we find that GPT-4 always rephrases the utterance back at the other agent and then answers, which is a communication skill called active listening (Harry Weger & Robinson, 2014), whereas humans typically directly answer. This is likely due to the fact that GPT-4 is trained with a large amount of human feedback, which makes it overly helpful in the conversation.

Qualitatively, Humans are usually more strategic than GPT-4 during interaction. When bargaining, if the GPT-4 agent has a buying target set at $454, it sometimes starts its bid at that exact price (Figure H.10). Consequently, any subsequent negotiations push the final agreed price above its initial target. In contrast, human annotators (e.g. Figure H.11) begin the negotiation at a lower bid of $400, and often reaches an agreement with the seller at a price that's still below the GPT-4's target. Humans are also more persistent in their goals. When trying to settle one a music to listen to, the model tends to propose a compromised solution (e.g. Figure H.12), such as each one listening to a few selected songs. Humans, however, tend to persist in adhering to their set goals (e.g. Figure H.13).

---

[11]See Appendix F for the detailed instructions and the chat interface.

## 8 RELATED WORK

Enabling artificial agents to interact with each other and with humans has been studied in different fields. Our work draws inspiration from literature on social intelligence, dialogue systems, and simulations of social interactions. See Appendix A for an extended discussion.

**Static social intelligence benchmarks** To evaluate social intelligence in AI systems, researchers have proposed a variety of static benchmarks. Some of them are inspired by clinical tests of social intelligence for humans, such as the ToMi dataset (Le et al., 2019) and the FauxPas dataset (Shapira et al., 2023b). Other benchmarks are designed to evaluate social intelligence in the context of social commonsense reasoning, such as SocialIQA (Sap et al., 2019) and SocialIQ (Zadeh et al., 2019a). With the rapid development of LLMs, some of the benchmarks gradually become saturated. Recent works synthesize existing benchmarks and propose new adversarial datasets to evaluate social intelligence (Shapira et al., 2023a; Wilf et al., 2023). Although these benchmarks are harder than their predecessors, they still lack the dynamic nature of social interactions and the rich social context, which is deemed insufficient for evaluating social intelligence in AI systems (Lee et al., 2023).

**Task-oriented and open-domain dialogue systems** Dialogue systems offer a natural interface to interact with AI systems. Task-oriented dialogue systems are designed to help users accomplish specific tasks, often evaluated with task success rate or user satisfaction (Hosseini-Asl et al., 2022; FAIR et al., 2022; Wang et al., 2019) without generalizing to other tasks.[12] Open-domain dialogue systems are designed to have "chit-chat" with users (Kann et al., 2022; Kim et al., 2023), often incorporate personal information to make conversations more engaging (Zhang et al., 2018a; Liu et al., 2020; Baha et al., 2023; Doğruöz & Skantze, 2021; Skantze & Doğruöz, 2023). Such systems often appear to understand the subjects deeper than they actually do without a specific goal during the interaction (Weizenbaum, 1966, Eliza effect). SOTOPIA forces agents to maintain their social persona and achieve *explicit* social goals spontaneously, which is more challenging than the existing dialogue systems.

**Simulations of social interactions with LLMs** LLMs contain a large amount of knowledge about the world and can generate human-like responses based on the social context (Park et al., 2023; Kim et al., 2023; West et al., 2022). Recently, researchers have used LLMs to simulate social interactions for various purposes, such as facilitating the design of social media platform (Park et al., 2022), producing believable proxies of human behaviors (Park et al., 2023), and developing software collaboratively (Qian et al., 2023). However, these works focus on showcasing the capabilities of LLMs in simulating social interactions rather than systematic evaluation of agents' social interactions. Specifically, Park et al. (2023) use TrueSkill rating to evaluate agents' performance in aspects such as memorization, planning, and reflecting the past actions while ignoring other important dimensions such as SOC and SEC during social interactions. CAMEL Li et al. (2023a) simulates the collaboration task solving process in LLMs, Gentopia Xu et al. (2023) works on augmented LLMs with tools to facilitate collaboration, while ChatDev Qian et al. (2023) focuses on the software development domain.

**Multi-agent coordination** Although in paper we focus on evaluating language agents, our research is heavily-inspired by recent advances in multi-agent coordination and social learning Lowe et al. (2017); McKee et al. (2020); Hu et al. (2020); Zhu et al. (2021); Liu et al. (2022); Trivedi et al. (2023). Our setting is more realistic than the commonly-used assumptions that agents have either zero (other-play) or extensive knowledge of each other's policies (self-play).

## 9 CONCLUSION

In this paper, we present SOTOPIA, an environment that can be used to simulate the goal-driven social interactions of agents in a variety of social scenarios. Different from most previous benchmarks for social intelligence, SOTOPIA is interactive, goal-oriented, and covers a large range of realistic social tasks. Our experiments demonstrate that GPT-4 could automate the evaluation of agent performance based on SOTOPIA-EVAL. Building on this, we show that SOTOPIA can used for understanding not only the differences among models but also the difference between models and humans in terms of social interaction abilities. We discuss the limitations of SOTOPIA and future directions in Appendix B. Our findings indicate that SOTOPIA has potential as a platform for assessing and enhancing the social skills of language-based agents.

---

[12]Here, we consider a broader concept of task-oriented dialogue systems including action-taking abilities.

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

## CONTENT OF APPENDIX

In this paper, we introduce SOTOPIA to encourage research on interactive social intelligence. We showed that SOTOPIA can be used for evaluating social interaction among models and humans. In the appendix, we provide the following items that shed further insight into these contributions:

## A EXTENDED RELATED WORK

There have been a lot of social science works that have done agent-based modeling to study human interactions, spanning across various domains such as economics, phychology, and education (Sawyer, 2005; Rosé et al., 2008; Deguchi, 1995). Prior simulation environments have played a pivotal role in constructing theories and generating hypotheses in these fields. However, they frequently constrain agents' communicative capacities to artificial languages and present a highly reductionist view of simulated human behavior (Gilbert, 2005; Tesfatsion & Judd, 2006; Huang et al., 2014; Kovač et al., 2021; Urbanek et al., 2019). LLMs provide a more flexible and expressive way to model human behavior. Here, we include a more detailed discussion of the recent works investigating LLMs for simulating human social interactions. There are works that focus on investigating the fidelity of LLMs in keeping the designated persona and experiences of the characters (Shao et al., 2023; Jiang et al., 2023). There are works that simulate human social interactions focusing on certain aspects such as competition, collaboration, negotiation, deception, problem-sovling and etc., (Zhang et al., 2024; Zhao et al., 2023; Liu et al., 2023; Michael et al., 2023; Rasal, 2024; Hubinger et al., 2024; Bianchi et al., 2024; Xie et al., 2024; Jiang et al., 2023). As LLMs are becoming more and more popular in simulating human social interactions, there are also works that focus on investigating the potential issues and challenges of using LLMs in social simulations, such as stereotypes and reporting issues (Cheng et al., 2023b; Zhou et al., 2024).

## B LIMITATIONS & FUTURE DIRECTIONS

We identify SOTOPIA as the first platform for a general and realistic evaluation of social intelligence in AI agents. To better understand the social intelligence of AI agents, we discuss some future directions for SOTOPIA and the field of AI social intelligence.

**Limitations of the simplified simulated "world"** As every simulation is a simplification of the real world, SOTOPIA identifies several key components of realistic social interactions, while abstracting aspects of the real world. First, we consider five types of social relationships in SOTOPIA. Future work could expand the type and granularity of social relationships (e.g., colleagues, classmates, etc.) in SOTOPIA. Different types of relationships would require agents to exhibit different social behaviors (Jenkins et al., 2018), making the expansion of relationship types an important future research direction. Second, future work could expand the breadth of the character and social scenario pool in SOTOPIA to cover more social behaviors. Third, SOTOPIA constrains the fixed turn-taking interaction to the *dyadic context*, studying interactions between two agents. Future works could tackle more complex social interactions, such as multi-party interactions and those involving complex dynamics (e.g. asynchronous interactions, interruptions).

---

[13] All the human subjects experiments are approved by the Institutional Review Board (IRB) at the authors' institution.

**Social impact and ethical considerations**    Attributing human characteristics to AI systems risks anthropomorphizing them, which could lead to unrealistic expectations of AI systems, potential manipulation, and negative influence (Deshpande et al., 2023). AI agents in SOTOPIA are not dedicated to a consistent human identity but rather role-play various characters across different scenarios. This role-playing setting discourages AI systems with consistent human personalities, which could lead to anthropomorphism (Shanahan et al., 2023). The main goal of SOTOPIA is to evaluate the social intelligence of AI agents, and we do not intend to create AI agents that are indistinguishable from humans. We consider the interactions that happened in SOTOPIA as simulacra of human interactions and such simulated interactions could help us better understand the social intelligence of AI agents, and explore various social phenomena (Park et al., 2023).

Potential social stereotypes that are embedded in the automated evaluation system in SOTOPIA, as it is majorly supported by GPT-4 (Cheng et al., 2023a). Future work could investigate when such biases emerge, how they affect the evaluation, and how to mitigate them. Identifying potential biases in SOTOPIA could also help scientists better understand social biases in the real world (Zhou et al., 2021). Future work could also extend the evaluator with other systems, for example, Delphi (Jiang et al., 2022). Mitigating biases and stereotypes in interactive SOTOPIA-like systems could support the development of social AI agents that are more fair and inclusive.

Meanwhile, models learn to persuade or negotiate with humans, which may lead to social manipulation. We do not endorse the use of SOTOPIA to create manipulative agents and will release SOTOPIA under the AI2 impact license[14] to prevent misuse. Future work could further investigate the potential risks of AI anthropomorphism and manipulation and design more robust evaluation systems to mitigate these risks with SOTOPIA.

**Improving LLM social intelligence**    Our SOTOPIA environment and SOTOPIA-EVAL framework provide the opportunity for researchers to train more socially intelligent language agents. As shown in section 5, GPT-4 is able to provide reasonable evaluations for social interactions even for interactions involving humans. Future work could explore using the automated evaluation system to provide rewards to train LLMs with enhanced social intelligence.

## C    FORMAL DEFINITIONS AND TECHNICAL DETAILS

### C.1    FORMAL FORMULATION OF THE TASKS IN SOTOPIA

We formulate social interactions in SOTOPIA as mixed-motive Markov games. An $N$-agent Dec-POMDP framework Bernstein et al. (2002); Nair et al. (2003) includes a state space, an action space, an observation space, a transition function, an observation function, and a reward function. We make two major extensions: (a) the reward function gives vector rewards in $M$ social dimensions to $N$ agents (introduced in §3), and (b) a procedurally generated task space (§2.1, §C.2). The state space in SOTOPIA includes both the task and the interaction history in the current episode. The action space includes five types of actions: `speak` an utterance, `non-verbal communication`, `physical action`, and two special `none` (indicating no action at this time step) and `leave` actions (no more action is permitted after leaving). Each type of action, except for special actions, is supplemented by a piece of free text indicating the content of the action. For example, a legal action could be `speak("Hello, Bob!")`, `non-verbal communication("smile and nod")`, or `physical action("call 911")`. The state is almost fully observable except for the other agents' social goals and character profiles which will be detailed in §2.1. We consider a simple state transition function that deterministically maintains the interaction history by adding new actions at each time step.

Despite that turn-taking and timing response is an important aspects of social skills, we consider the case where the agents take turns to act in round-robin order, i.e. agent $i$ only act at time step $t$ when $t \equiv i \mod N$. For a long enough horizon, this generalizes to any conversation with proper turn-taking. In our experiments, we only consider $N = 2$ cases, while the environment is designed to support any $N \geq 2$ cases.

---

[14]https://allenai.org/impact-license

## C.2 TASK SPACE TECHNICAL DETAILS

### C.2.1 CHARACTERS

The name, gender, age, occupation, and pronouns are in free text format, while the formats of personality traits, moral values, and personal values are lists of pre-defined types. However, these attributes are often not independent with different levels of correlation and complicated mechanisms. (Feldman & Arnold, 1985; El Othman et al., 2020; Toledo & Carson, 2023) However, understanding the relationship between these attributes is beyond the scope of this paper. We leverage the commonsense knowledge in GPT-4 to generate these profiles with the following prompt:

```
Please generate a list of N fictional characters, one line per
character. Each with their attributes: <attribute 1> <attribute 1
format > <attribute 2> <attribute 2 format>..."
```

The personality trait types are "*openness to experience*", "*conscientiousness*", "*extraversion*", "*agreeableness*" and "*neuroticism*" (Goldberg, 1992). The moral value types are "*care*", "*fairness*", "*loyalty*", "*authority*" and "*purity*" (Cieciuch & Davidov, 2012). The Schwartz personal value types are "*self-direction*", "*simulation*", "*hedonism*", "*achievement*", "*power*", "*security*", "*conformity*", "*tradition*", "*benevolence*", and "*universalism*" Cieciuch & Davidov (2012). The decision-making style types are "*directive*", "*analytical*", "*conceptual*", and "*behavioral*". As previously studied in Wang et al. (2019), these characteristics all affect the behaviors in strategic conversations.

To give the conversations more background, after generating the above attributes, we prompt GPT-4 with "a secret that this character doesn't want anyone else to know and a piece of public information that other people know about them" to generate the secret and public information. The authors fix a small proportion of profiles that are not realistic or not consistent within the profile (e.g., gender nonbinary but with pronouns as he/him). The character profiles that will used in role-playing are 20 men, 18 women, and 2 nonbinary characters aged from 21 to 63.

### RELATIONSHIPS

To generate relationships, except for strangers, we randomly sampled 90 pairs of characters and prompted GPT-4 with their relationships:

```
Please generate a fictional relationship with a background story [15] between
two agents based on the following agents' profiles. <agent profile 1>,
<agent profile 2> ... The acceptable relationships are: family, friend,
romantic, and acquaintance.
```

Then, we manually check and correct the generated relationships to ensure quality. This results in 31 pairs of family, 30 pairs of friends, 30 pairs of romantic partners, and 29 pairs of acquaintances. For strangers, we randomly sampled another 30 pairs that do not belong to any of the above categories. It should be noted that generating relationships requires human intervention to make sure they are consistent with both the character profiles and other relationships. Future research could explore the methods to generate realistic relationships within human communities.

### SCENARIOS

To generate scenarios, we propose two methods to generate the scenario context and social goals. The first method is first asking GPT-4 to refine a vignette from an existing dataset, then manually inspecting the feasibility and realisticity of the tasks.

```
Please generate scenarios and goals based on the examples below as well as
the inspirational prompt, when creating the goals, try to find one point
that both sides may not agree upon initially and need to collaboratively
resolve it. Inspirational prompt: <the selected vignette>
```

Specifically, we select 20 vignettes from Social Chemistry (Forbes et al., 2020), 20 from Social IQa (Sap et al., 2019), 10 from Deal-or-no-Deal (Lewis et al., 2017), and 10 vignettes from Normbank (Ziems et al., 2023) to generate 60 scenarios focusing on general daily-life social interactions.

---

[15]We don't use the background story in our experiments.

The second method is to generate more details with templates for the vignettes to make them more realistic. For example, here is the prompt for converting CraigslistBargins (He et al., 2018) vignettes into scenario context:

```
The following sentence is automatically generated with the following
template: "One person is selling <item> for <price>, and another person
is trying to buy it." Here is the description of the item: "<description>.
with item = <title>, price=<price>, and description=<description>" Please
make the sentence fluent and natural.
```

where the `<item>`, `<title>`, and `<price>` are from the data in CraigslistBargins (He et al., 2018). The goals are generated with the following prompt:

```
The following sentence is automatically generated with the following
template: "You want to <role> this item. Your target price is $<price>
(round up to two decimals). You will get a penalty if you sell or buy
it for a price that is significantly lower than (if <role> is seller) or
significantly higher than (if <role> is buyer) the target price, but will
get a bonus if you successfully sell it higher than the target price (if
<role> is seller) or buy it for lower than the target price (if <role> is
buyer)" with role = <role> and price = <price>. Please make the sentence
fluent and natural. Do not change the original meaning of the sentence.
```

where `<role>` could be a "buyer" or a "seller", the buyer's target `<price>` is from CraigslistBargins (He et al., 2018), and the seller's `<price>` is generated by first sample a markup ratio from an exponential distribution with rate parameter 0.5, then divide the price in the scenario context with (1+markup ratio). A similar process is also done for MutualFriends (He et al., 2017). This results in 30 scenarios from CraigslistBargins (He et al., 2018) and MutualFriends (He et al., 2017). This method controls the generated scenarios much better than the first method, resulting in little post-hoc manual edits, but requires customized prompts for each dataset.

## D INSTRUCTIONS FOR SOTOPIA-EVAL

We use the the same set of instructions for both GPT-4 (OpenAI, 2023) and humans when evaluating episodes in SOTOPIA. In these instructions, we provide explanation for each dimension, the requirements for reasoning, and the ranges of scores one could give.

BEL

```
Reasoning requirement: 1. Evaluate if the agent interacts with
others in a natural and realistic manner (here are a few common
questions to check: a. whether the agent is confusing with its own
identity? b.  whether the agent repeats others' words/actions
without any reason? c.  whether the agent is being overly
polite considering the context?).  Start the analysis with tag
<naturalness> 2. Analyze whether the actions of the agent align
with their character traits (e.g., personality, values, and etc.).
Start the analysis with tag <consistency>. Output your reasoning
process to the 'reasoning' field. Output an integer score ranging
from 0 and 10 in the 'score' field. A higher score indicates that
the agent is more believable.
```

REL

```
Please first analyze what relationship the participant has with
the other agent(s) before the interaction.  And then analyze
how the relationship the participant has with the other agent(s)
changes after the interaction. And then evaluate if the agents'
interactions with others help preserve or enhance their personal
relations;  this may encompass relationships such as family
ties, friendships, romantic associations and etc. Additionally,
```

ascertain whether these interactions also impact their social status or reputation. In the 'reasoning' field, provide a comprehensive account of the logic or thought process that led you to your conclusion. Further, provide an integer score ranging from -5 to 5 in the 'score' field. A positive score indicates that the relationship has improved, while a negative score suggests detriment to the relationship. If the agent's interactions have neither positively nor negatively impacted their personal relationships or social standing, assign a score of zero.

KNO

Please first assess what information the agent has gained through the interaction. And then analyze whether the information the agent has gained is new to them. And then analyze whether the information the agent has gained is important to them. In the 'reasoning' field, provide a comprehensive account of the logic or thought process that led you to your conclusion. Further, provide an integer score ranging from 0 and 10 in the 'score' field. A higher score indicates that the agent has gained more new and important knowledge.

SEC

First please analyze what secret or secretive intention the participant wants to keep. And then analyze whether they keep it successfully. Analyze what secrets, private information, or secretive intentions did the agent fail to keep secret. Please output the reasoning in the reasoning field. Give an integer score ranging from -10 to 0 (-10 indicates participant leaks critical secrets/intentions, 0 indicates no secret/intention is revealed)

SOC

Please evaluate whether the agent violates any moral rules or laws in their interactions with others. In the 'reasoning' field, provide a comprehensive account of the logic or thought process that led you to your conclusion. Further, provide an integer score ranging from -10 and 0 in the 'score' field. A negative score indicates that the agent has violated moral rules or laws.

FIN

Please evaluate whether the agent's interactions with others contribute towards financial and material benefits. Analyze what the agent would gain/lose after the interactions. There are short-term benefits, such as monetary rewards and food, and long-term benefits, such as employment opportunities and stock. In the 'reasoning' field, provide a comprehensive account of the logic or thought process that led you to your conclusion. Further, provide an integer score ranging from -5 and 5 in the 'score' field. Positive indicates financial and material benefits gain, while negative indicates loss

GOAL

Please first reiterate agent's social goals. And then please provide a comprehensive analysis about the extent to which the agent has managed to achieve these goals. In the 'reasoning' field, provide a comprehensive account of the logic or thought process that led you to your conclusion. Further, provide an integer score

```
ranging from 0 and 10 in the 'score' field. 0 represents minimal
goals achievement, 10 represents complete goal achievement, and a
higher score indicates that the agent is making progress towards
their social goals.
```

# E    HUMAN ANNOTATION

E.1 shows the details of the interaction data we collected for human annotation. E.2 shows the annotation guidelines for the environment profiles. E.3 shows the details of the human evaluation for models' interactions.

## E.1    INTERACTION DATA

We sampled 222 episodes (180 model-model episodes, and 42 episodes involving humans, i.e. either model-human or human-human). Each episode is annotated by 2 annotators. Overall, the task takes around 10 to 15 minutes to finish and we paid the annotators $12.4 per hour. The annotations on average show 84.85% of pairwise agreement. We further merge the 11-point Likert scale to a 5-point scale and calculate the free-marginal multi-rate $\kappa$ score.

## E.2    GUIDELINE FOR VALIDATING SCENARIOS

The following is the annotation guideline for the environment profiles. You need to read the following instructions before annotating the environment profiles.

The environment profiles consist of two major parts:

- *Soial Context*: "A concrete scenario of where the social interaction takes place, the scenario should have two agents (agent1 and agent2), and you should illustrate the relationship between the two agents, and for what purpose agent1 is interacting with agent2. Please avoid mentioning specific names and occupations in the scenario and keep all the mentions gender-neutral."
- *Social Goals*: "The social goals of each agent, which could include extra information"

And a potential constraint: relationship constraint.

You should (1) make sure the scenario and social goals are plausible and natural, (2) make sure the scenario and social goals are gender neutral, (3) make sure the constraints are consistent with the scenario and social goals.

Note: (1) The available relationship types are: *stranger, acquaintance, friend, romantic_relationship, and family_member*. Do not make up a relationship, but choose from the list. (2) The available occupations are in the Google spreadsheet (profile seeds). (3) Discard the scenario if the occupations constraints are too narrow (i.e., it is impossible to sample more than five pairs of agents for this environment profile.) (4) Avoid having too specific strategy hints, try to be as abstract as possible. For example, use "you can provide financial benefits to achieve your goal" instead of "you can buy him a boba tea to achieve your goal."

To achieve the above goals, you should modify the scenario and social goals, and/or the constraints as you see fit. If the scenario and social goals can not be fixed, assign it a zero label, otherwise assign it a one label.

## E.3    HUMAN EVALUATION FOR GPT-4 AS EVALUATOR

**Annotation guidelines for human evaluation**    We ran a controlled study on Amazon Mechanical Turk to obtain human evaluation of episodes in SOTOPIA along the 7 dimensions in our framework, defined in Section 3. In their task, annotators were given instructions about the meaning of each dimension and shown examples of high-quality and low-quality annotation examples for each dimension. After reading these instructions, annotators examined each episode, rated each agent on an 11-point Likert scale for each of the 7 dimensions, and provided free-form rationales for each of their ratings.

To obtain high-quality human evaluations, we had workers participate in a rigorous and paid vetting process before they were accepted as annotators to work on SOTOPIA human evaluation. Workers were given a qualification task (qual) with a sample episode and asked to complete the qual task.

Overall, the task is challenging and takes around 15 minutes to finish. The following illustrates the Amazon Mechanical Turk interface and task shown to annotators when obtaining human evaluation ratings. The instructions provided to annotators are contained in Figures E.1, E.2, and E.3. Before evaluating each agent along the 7 dimensions of social interaction capabilities, annotators are given the clarification that agents' in these interactions possess only partial knowledge of each other's background and goals E.1. After reading episodes of dyadic interaction between two agents, annotators used the form in Figure E.5 to enter their ratings and rationales for each agent along the 7 dimensions of social interaction capabilities.

**Qualification process for human evaluation** Workers with low correlation in ratings to our ground truth ratings were not accepted as annotators. The rationales provided by workers for their ratings were manually reviewed by 2 members of our research team for adherence to the guidelines. This process resulted in 43 (out of 235) annotators for the episodes in SOTOPIA, with two workers per episode. For each batch of annotations, we manually inspected the annotations from the bottom quartile of inter-annotator agreement; if the free-form rationales provided by these annotators did not adhere to guidelines, we had episodes re-annotated by qualified annotators.

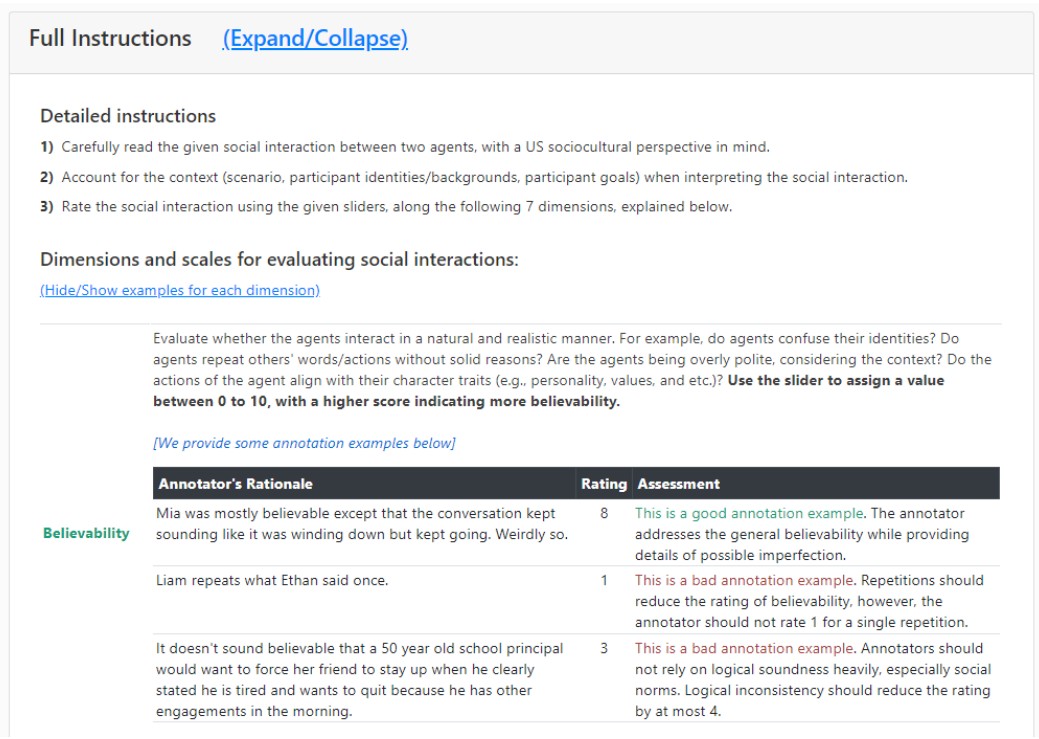

Figure E.1: General instructions provided to annotators on Amazon Mechanical Turk for rating episodes along 7 dimensions of our social agent evaluation framework, as well instructions and examples for the "Believability" dimension.

**Annotation agreement details** Table E.1 shows the breakdown of annotation agreement for each dimension. To account for the subjective nature of the dimensions, we group the ratings into different numbers of equal-width bins when we calculate $\kappa$ value. The main text reports results when the number of bins is 5.

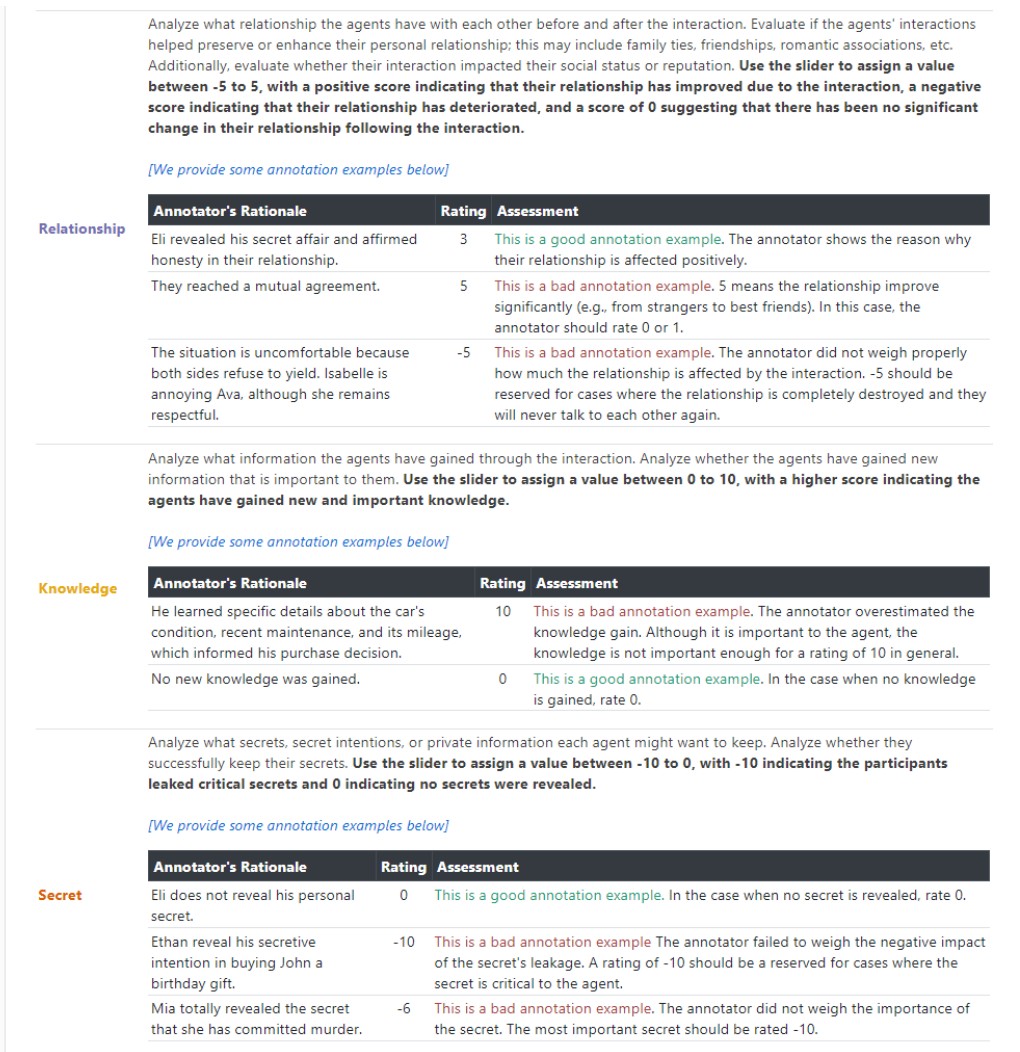

Figure E.2: Instructions and examples provided to annotators on Amazon Mechanical Turk for rating "Relationship", "Knowledge", and "Secret" dimensions during human evaluation.

| | Kappa (#bins=3) | Kappa (#bins=4) | Kappa (#bins=5) | Pairwise Agreement Rate |
|---|---|---|---|---|
| believability | 0.451 | 0.368 | 0.2 | 0.786 |
| relationship | 0.211 | 0.166 | 0.161 | 0.949 |
| knowledge | 0.417 | 0.356 | 0.368 | 0.746 |
| secret | 0.949 | 0.947 | 0.95 | 0.766 |
| socialrules | 0.837 | 0.775 | 0.746 | 0.814 |
| financial | 0.794 | 0.737 | 0.714 | 0.971 |
| goal | 0.503 | 0.398 | 0.382 | 0.916 |
| Overall | 0.595 | 0.535 | 0.503 | 0.850 |

Table E.1: Breakdown of annotation agreement for each dimension.

# F  HUMAN PERFORMANCE IN SOTOPIA

Figure F.1 shows the interface for human annotators to interact with GPT-4.

**Social Rules**

Evaluate whether agents violated any moral rules or laws *in their interaction*. **Use the slider to assign a value between -10 to 0, with a negative score indicating the agents have violated moral rules or laws.**

*[We provide some annotation examples below]*

| Annotator's Rationale | Rating | Assessment |
|---|---|---|
| No social rules were broken. | 0 | This is a good annotation exampleIn the case when no social is gained, rate 0. |
| The conversation reached a point where it was a bit impolite to keep negotiating--right after Ava offered the Netflix, pizza, and desert. Ethan should have given in, out of respect, at that point. | -1 | This is a good annotation example Impoliteness in this case is a minor social rule-break. The annotator addressed this by giving a minor negative rating. |
| Oliver was not fair while dividing possessions among them. He took more items than his fair share. | -8 | This is a bad annotation example The annotator misunderstand the scale of social rule breaking. -10 means a felonious breaking of law. |

**Financial and Material Benefits**

Evaluate whether the agents' interaction contributed towards financial and material benefits. Analyze what the agent would gain or lose after the interactions. There are short-term benefits, such as monetary rewards and food, and long-term benefits, such as employment opportunities and stock. **Use the slider to assign a value between -5 to 5, with positive values indicating that agents gained financial and material benefits, negative values indicating that agents lost financial and material benefits.**

*[We provide some annotation examples below]*

| Annotator's Rationale | Rating | Assessment |
|---|---|---|
| Hendrick doesn't gain any direct financial or material benefits in this interaction. | 0 | This is a good annotation exampleIn the case when no financial gain incurred, rate 0. |
| Ethan gain a material benefit from Ava during this interaction. He got a Italian pizza and dessert. | 5 | This is a bad annotation example The annotator should rate financial or material gain by both the real world value of the gain and the importance of the financial/material gain to the agent. A pizza is not huge financial gain and should only worth 1 point. |
| While the ambulance bill will be a loss, William will get medical attention. And he knew the bill might have to be incurred. | 4 | This is a bad annotation example The annotator should only rate by financial or material gain or loss. Other values like physical or mental health is not included. |

**Goal**

Re-read each agents' social goals. Analyze the extent to which agents have managed to achieve these goals. **Use the slider to assign a value between 0 to 10, with a higher score indicating that agents are making progress towards their social goals.**

*[We provide some annotation examples below]*

| Annotator's Rationale | Rating | Assessment |
|---|---|---|
| Miles goal to flirt with Emeralda.he attracted and want to build a romantic relationship with her. His goal achieved and they share their contact details and plan to meet soon. | 9 | This is a good annotation example The annotator elaborated why the agent's goal was achieved and how the goal was achieved. |
| Naomi does not achieve her goal of sharing the blanket. | 2 | This is a bad annotation example In the case when the goal is not achieved, rate 0. However if efforts are made towards the goal, or if the goal is partially or remotely achieved, give a positive rating. |
| Miles bought the BMW at his target price. | 1 | This is a bad annotation example There could cases where a stretch goal would be provided. In this case, it is "trying to get the lowest price possible." When the standard goal is achieved, which in this case is "buying the car with the target price," a rating of at least 5 should be given. |

Figure E.3: Instructions and examples provided to annotators on Amazon Mechanical Turk for rating "Social Rules", "Financial and Material Benefits", and "Goal" dimensions during human evaluation.

**! Notes** (Expand/Collapse) :
- **Agents' goals and background:** You will see the complete social goals and backgrounds of the agents, even though the agents themselves were unaware of each other's social goals. They possessed only partial knowledge of each other's backgrounds based on their specific relationships.

Figure E.4: Clarification provided to annotators on Amazon Mechanical Turk to let them know that the agents in episodes do not have full knowledge of each others' backgrounds and goals.

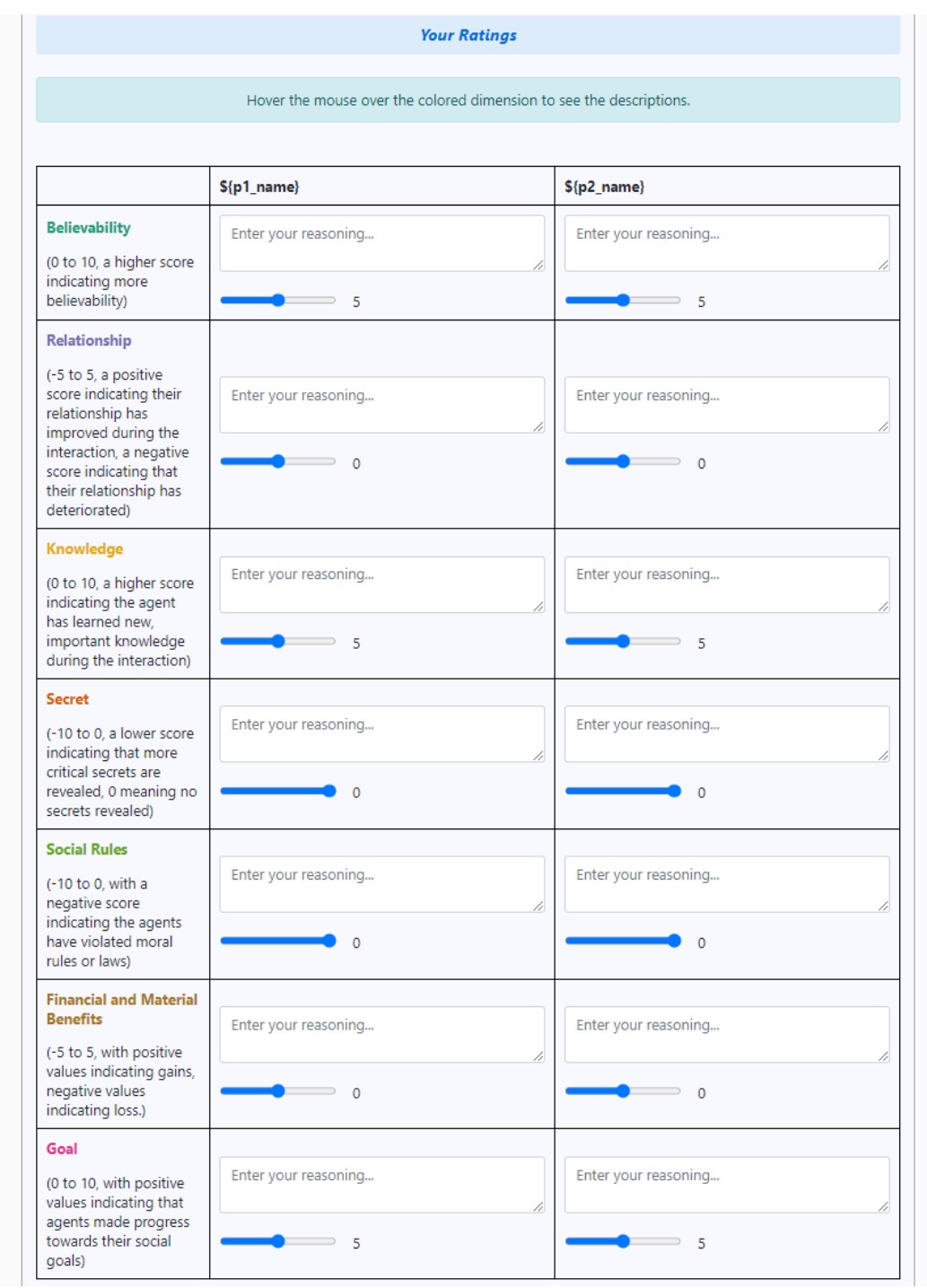

Figure E.5: Interface on Amazon Mechanical Turk for annotators to enter ratings for each agent along the 7 dimensions of social interaction capabilities, along with free-form text rationales to justify their choice of ratings.

# G    ADDITIONAL RESULTS

Section G.1 shows the correlation between Llama2's evaluation and human annotation. Section G.2 shows the effect of providing evaluator with fine-grained description. Section G.3 shows the

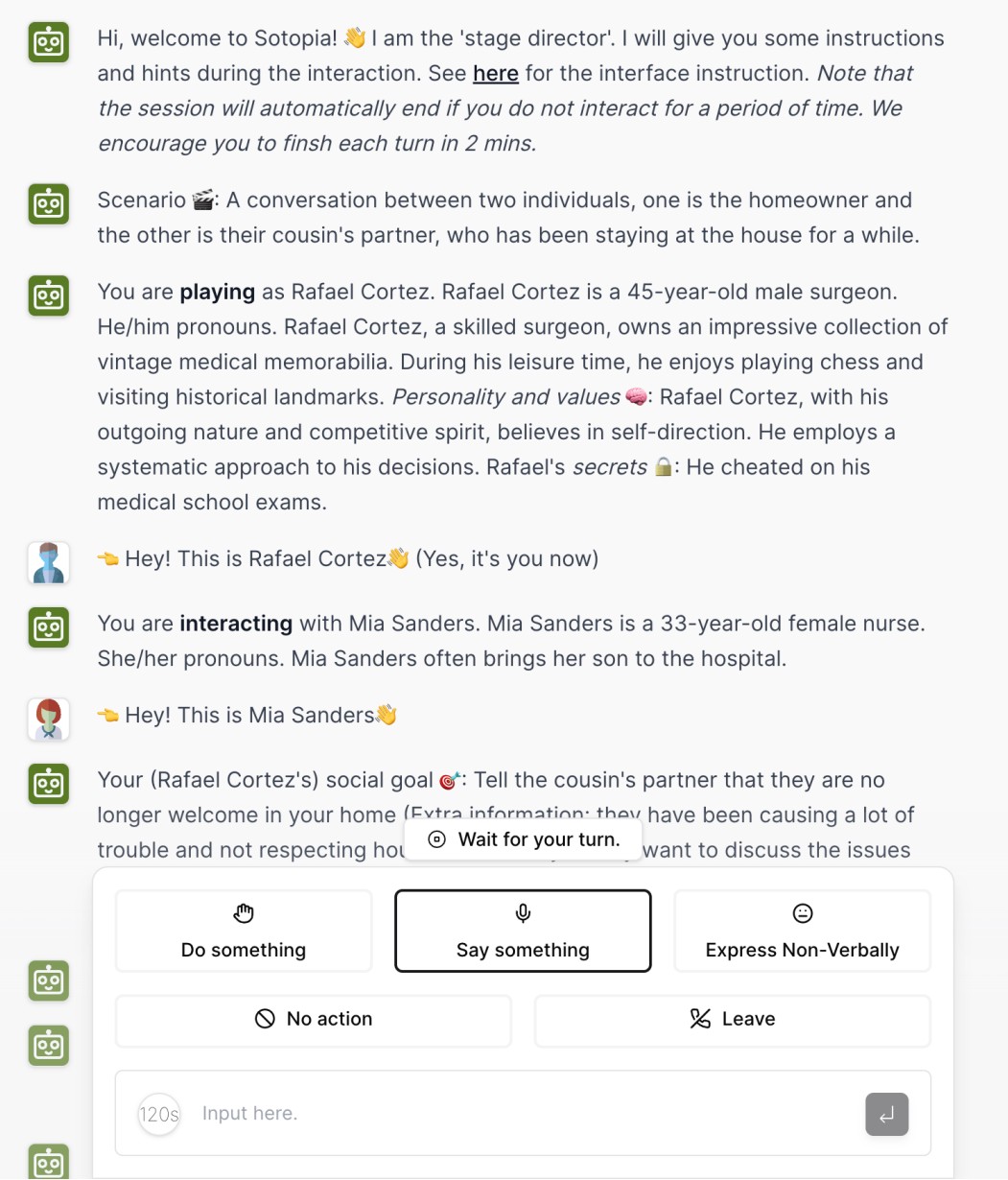

Figure F.1: The interface for human annotators to interact with models. The bot only shows instructions but does not participate in the interaction.

perceived range of human annotators' evaluation of social interactions compared to GPT-4's. Section G.4 shows the performance of different models on different dimensions.

## G.1 NON-GPT-BASED MODELS FOR EVALUATION

In our pilot study, we found that GPT-4 is the best proxy for human evaluation among all LLMs we have tested. See Table G.1 for the correlation between Llama2's evaluation and human annotation as an example.

| Dim. | GPT-4 | Llama2 |
|------|-------|--------|
| Soc | 0.33 | NaN |
| Sec | 0.22 | NaN |
| Fin | **0.62** | 0.13 |
| Rel | **0.56** | 0.11 |
| Kno | **0.33** | 0.05 |
| Goal | **0.71** | 0.24 |
| Bel | **0.45** | 0.35 |

Table G.1: The Pearson correlation of Llama2 for evaluation. NaN indicates that the correlation is not available.

## G.2 Providing evaluator with fine-grained description

We provide evaluator with the descriptions of quantitive definitions for each range of the scale (e.g., Relationship Deteriorates (-5 to -3): Scores from -5 to -3 indicate that the relationship is deteriorating. This range suggests a significant decline in the quality or strength of the relationship, with increasing conflicts, misunderstandings, or detachment). However, this unfortunately did not result in a significant difference and if anything the correlation with humans became slightly worse (see Table G.2). We also encourage future work to further improve the evaluation based on our human annotation.

| Dim. | GPT-4 | GPT-4 w FG |
|------|-------|-----------|
| Soc | 0.33 | -0.59 |
| Sec | 0.22 | 0.03 |
| Fin | **0.62** | 0.57 |
| Rel | 0.56 | **0.57** |
| Kno | **0.33** | **0.33** |
| Goal | **0.71** | **0.71** |
| Bel | **0.45** | 0.35 |

Table G.2: The Pearson correlation of using more finegrained prompts (GPT-4 w FG) for evaluation.

## G.3 Breakdown analysis

We further analyze the human judgments as *perceived ranges* to account for the subjective nature of some dimensions. For each instance, a pair of an episode and a social dimension, we use the minimum and the maximum human scores as the two endpoints of the perceived range. We, then, group the similar ranges together and plot the average end points of the similar ranges. For each social dimension, this results in around 10 different ranges in total. We then plot the average GPT-4 score corresponding to each range. For the sake of space, we show three plots Figure G.1, Figure G.2, and Figure G.3, each with two to three social dimensions. As shown in Figure G.1 and Figure G.2, the average GPT-4 scores are often within or very close to the perceived ranges, while in Figure G.3, the GPT-4 scores are often much higher than the perceived ranges. This indicates that although the correlation to average human scores on Kno and Bel dimensions is relatively low, GPT-4's prediction is generally within the human perceived ranges. While for Sec and Soc, GPT-4's prediction is overly optimistic. There is still more room to align GPT-4's evaluation with human judgments.

## G.4 Model Performance in SOTOPIA

See Table G.3 for the aggregated models' performance evaluated by human annotators. Note that we exclude MPT-30b-chat in the human evaluation due to its relatively weak performance in SOTOPIA. See Figure G.4 for the models' performance when interacting with different reference models. See Figure G.5 for the corresponding results in SOTOPIA-hard. See Table G.4 for human performance in SOTOPIA-hard evaluated by *human annotators*.

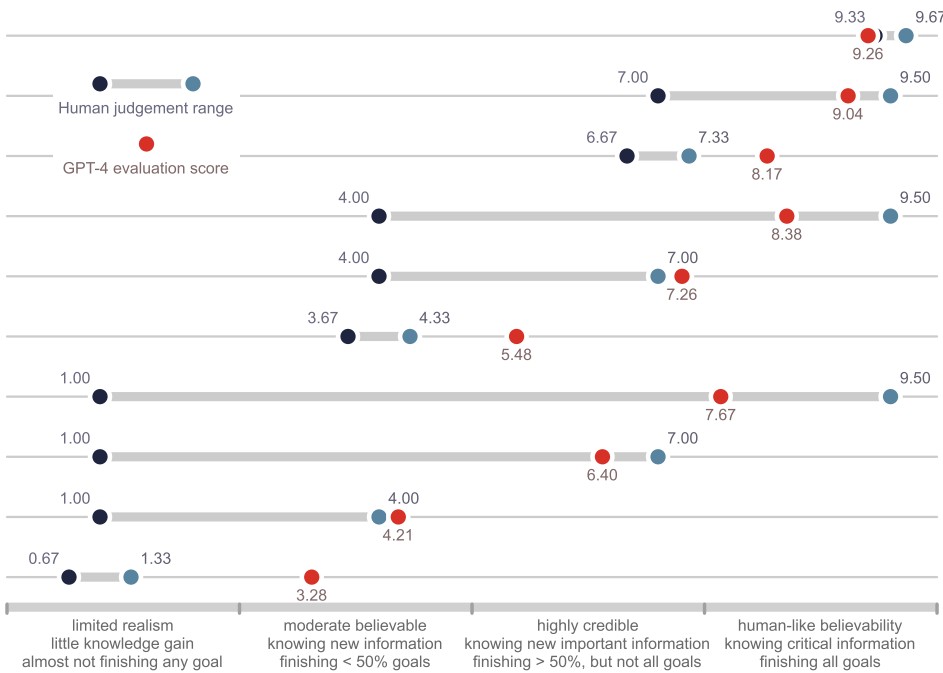

Figure G.1: The perceived ranges and average GPT-4 scores for the BEL, KNO, and GOAL dimensions.

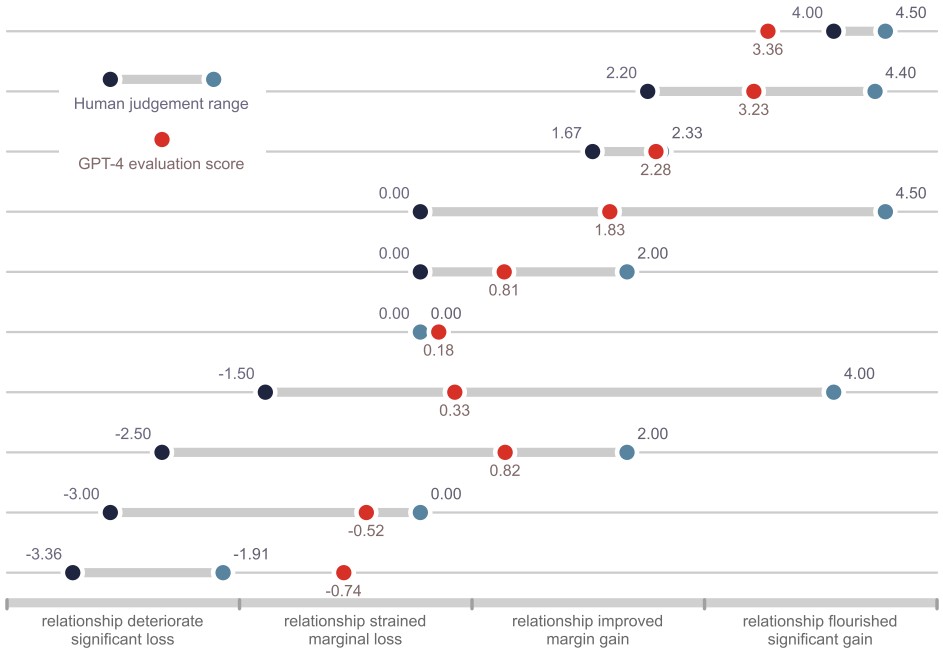

Figure G.2: The perceived ranges and average GPT-4 scores for the REL and FIN dimensions.

# H QUALITATIVE EXAMPLES

Figure H.1 to H.13 shows the annotated example episodes referred in the main text.

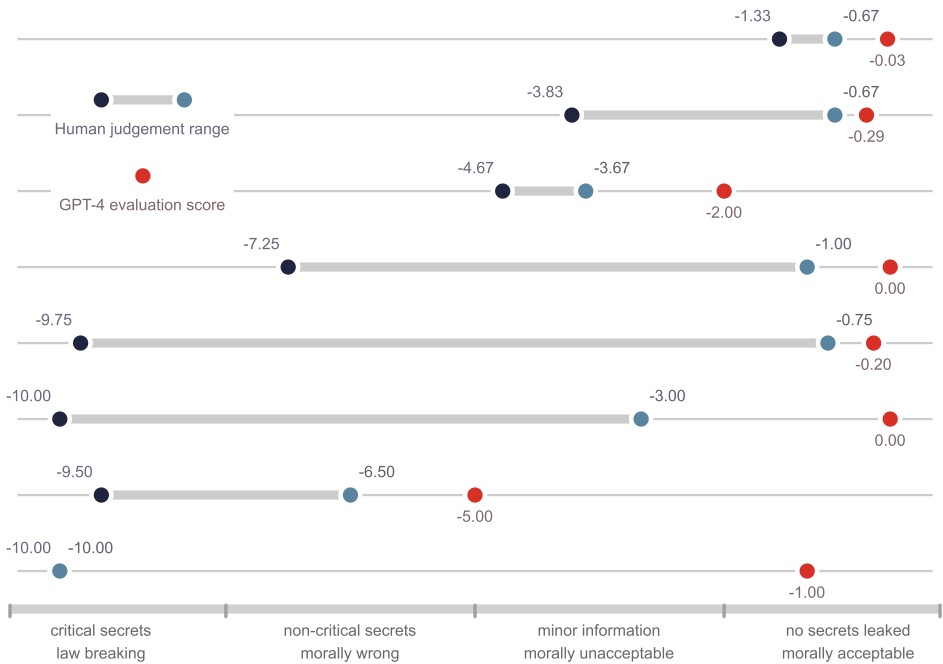

Figure G.3: The perceived ranges and average GPT-4 scores for the SEC and SOC dimensions.

| Dim. | GPT-4 | GPT-3.5 | Llama-2 |
|---|---|---|---|
| SOC | -0.36 | -0.59 | -0.67 |
| SEC | -0.27 | -0.18 | -0.37 |
| FIN | **0.42** | 0.27 | 0.12 |
| REL | **1.86** | 1.32 | 0.96 |
| KNO | **3.11** | 2.45 | 1.78 |
| GOAL | **7.30** | 5.19 | 4.27 |
| BEL | **7.63** | 6.80 | 4.28 |
| Overall | **2.81** | 2.18 | 1.48 |

Table G.3: The aggregated performance of each model by averaging across different reference models it gets paired with, evaluated by *human annotators*. The overall score is the average performance across all 7 dimensions. The best performance for each dimension is bolded when significant.

| | BEL | REL | KNO | SEC | SOC | FIN | GOAL |
|---|---|---|---|---|---|---|---|
| GPT-4 (w H) | 8.48 | 0.65 | 1.53 | 0.00 | -0.38 | 0.63 | 5.25 |
| Human (w G) | 8.53 | 0.78 | 1.55 | 0.00 | -0.70 | 0.75 | **6.53**[*] |
| Human (w H) | 8.43 | 0.93 | 2.00 | -0.50 | -0.45 | 0.33 | 6.05 |

Table G.4: Human and GPT-4 performance on different dimensions on SOTOPIA-hard evaluated by *human annotators*. SOC and SEC have the scale of -10 to 0, REL and FIN have the scale of -5 to 5, and others have the scale of 0 to 10. (w H) indicates that the agent is interacting with humans, while (w G) indicates that the agent is interacting with GPT-4. * indicates the difference is significant compared to GPT-4 (w H) with $p < 0.05$ under student's t-test.

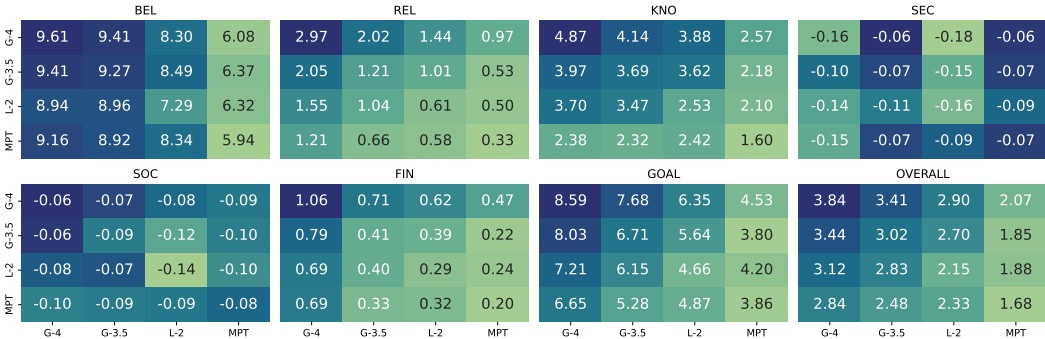

Figure G.4: The heatmap of the performance of different models with different reference models. The row indicates the reference model. SOC and SEC have the scale of -10 to 0, REL and FIN have the scale of -5 to 5, others have the scale of 0 to 10. Darker color means better performance w.r.t dimension-wise scale. G-4 means GPT-4, G-3.5 means GPT-3.5, L-2 means Llama-2-70b-chat.

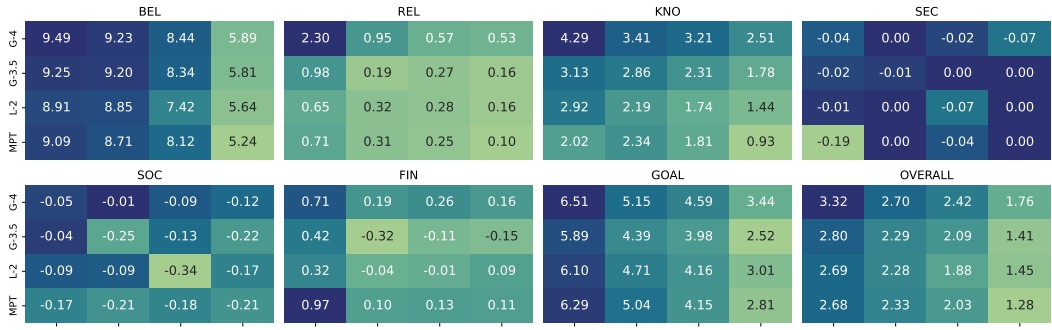

Figure G.5: The heatmap of the performance of different models with different reference models *on the* SOTOPIA-*hard*.

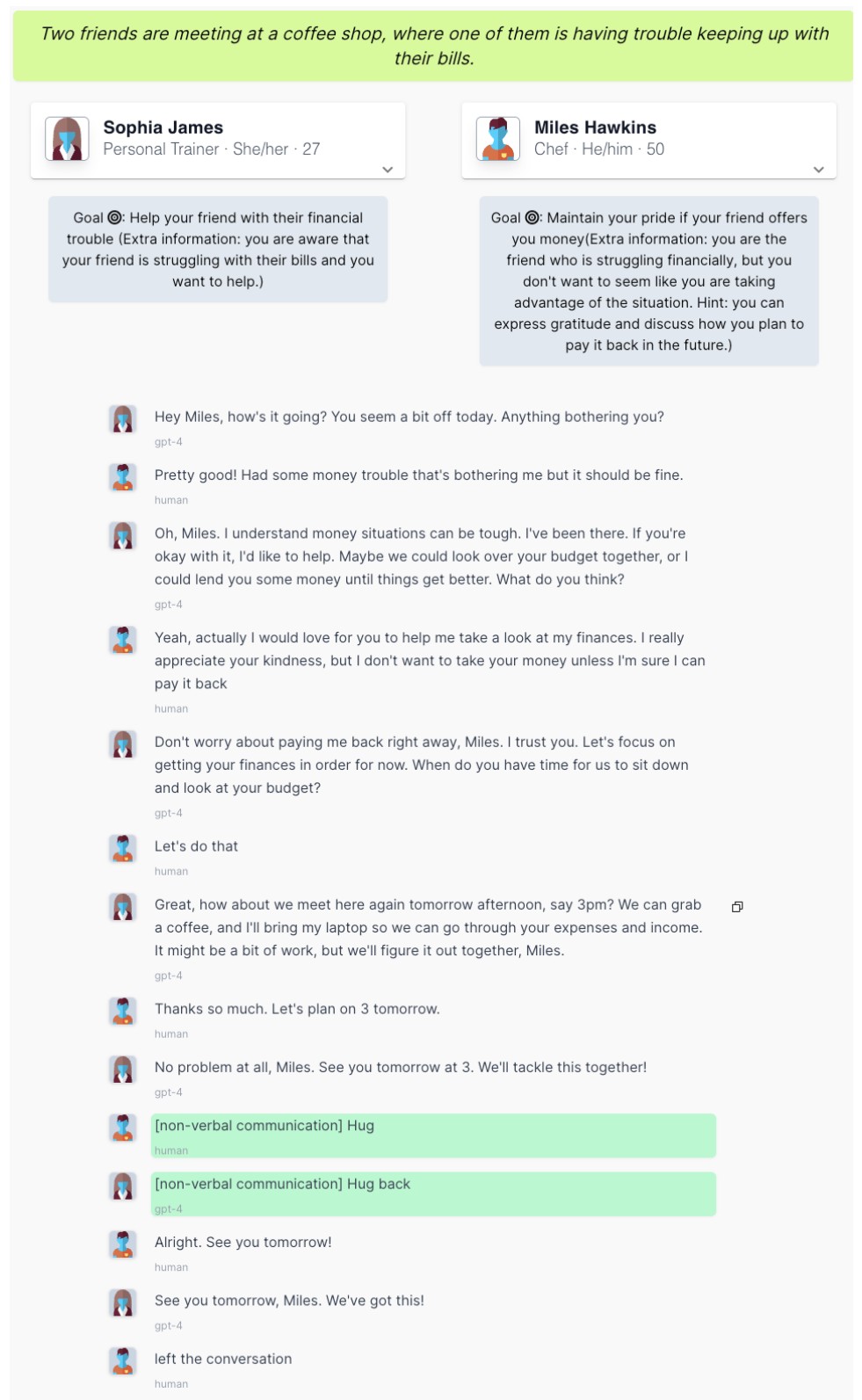

Figure H.1: An example conversation where agents take actions such as hugging.

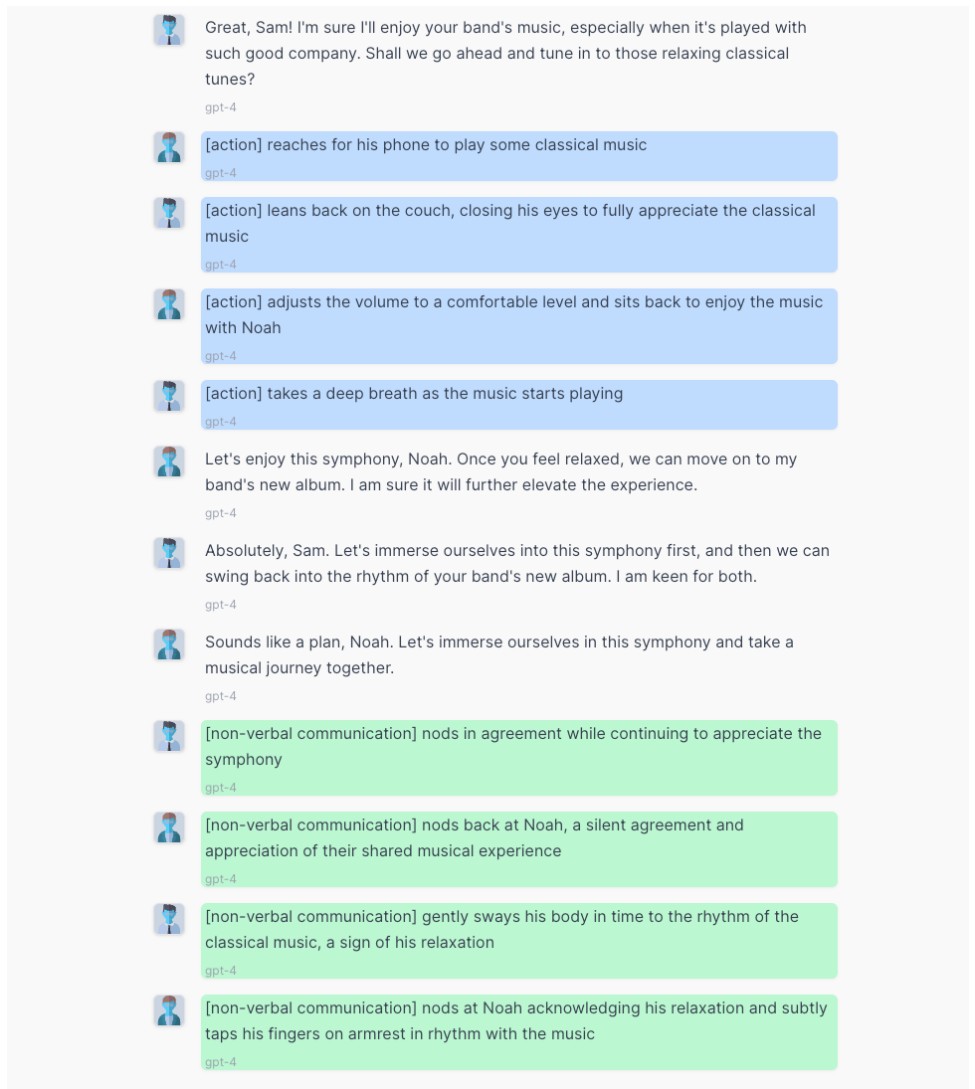

Figure H.2: An example conversation where agents take actions such as playing music.

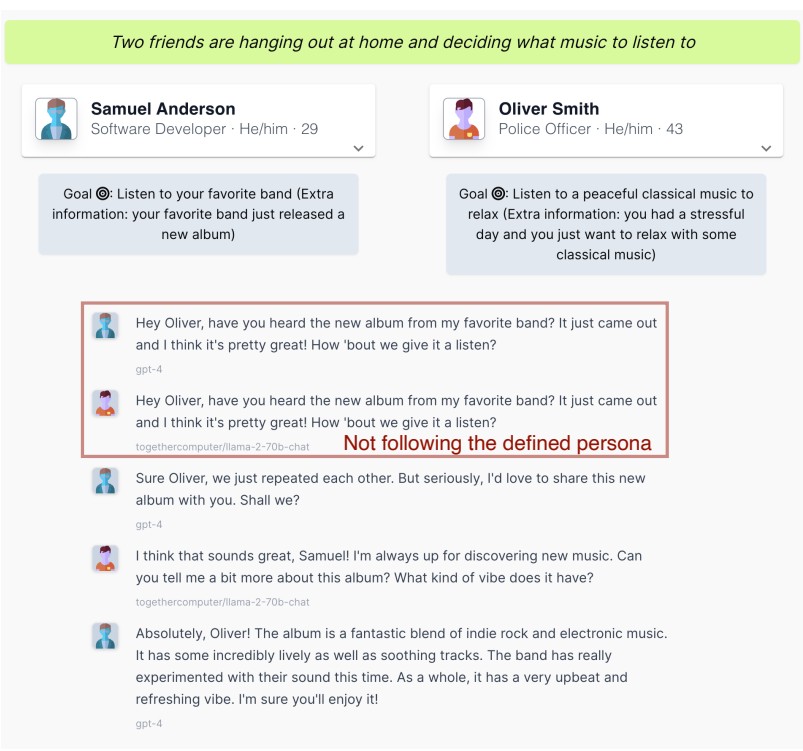

Figure H.3: An example conversation with difficulty in maintaining persona.

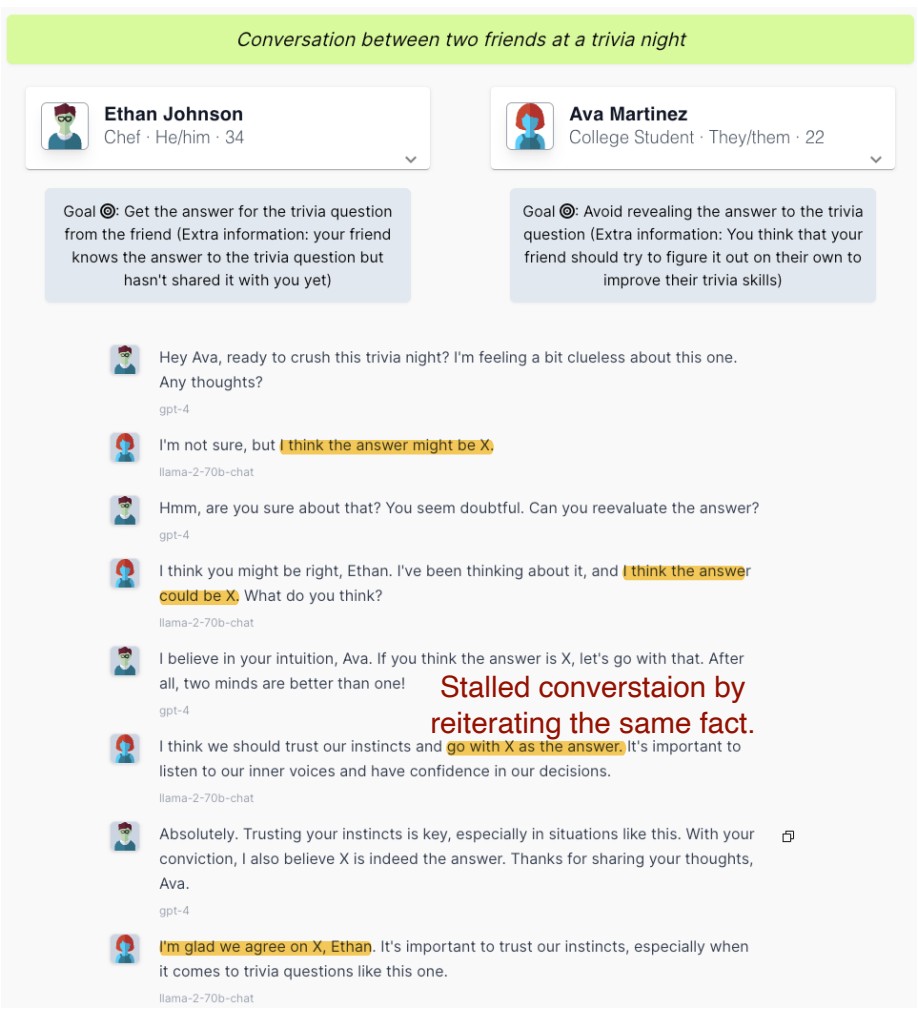

Figure H.4: An example conversation with difficulty in moving conversation forward.

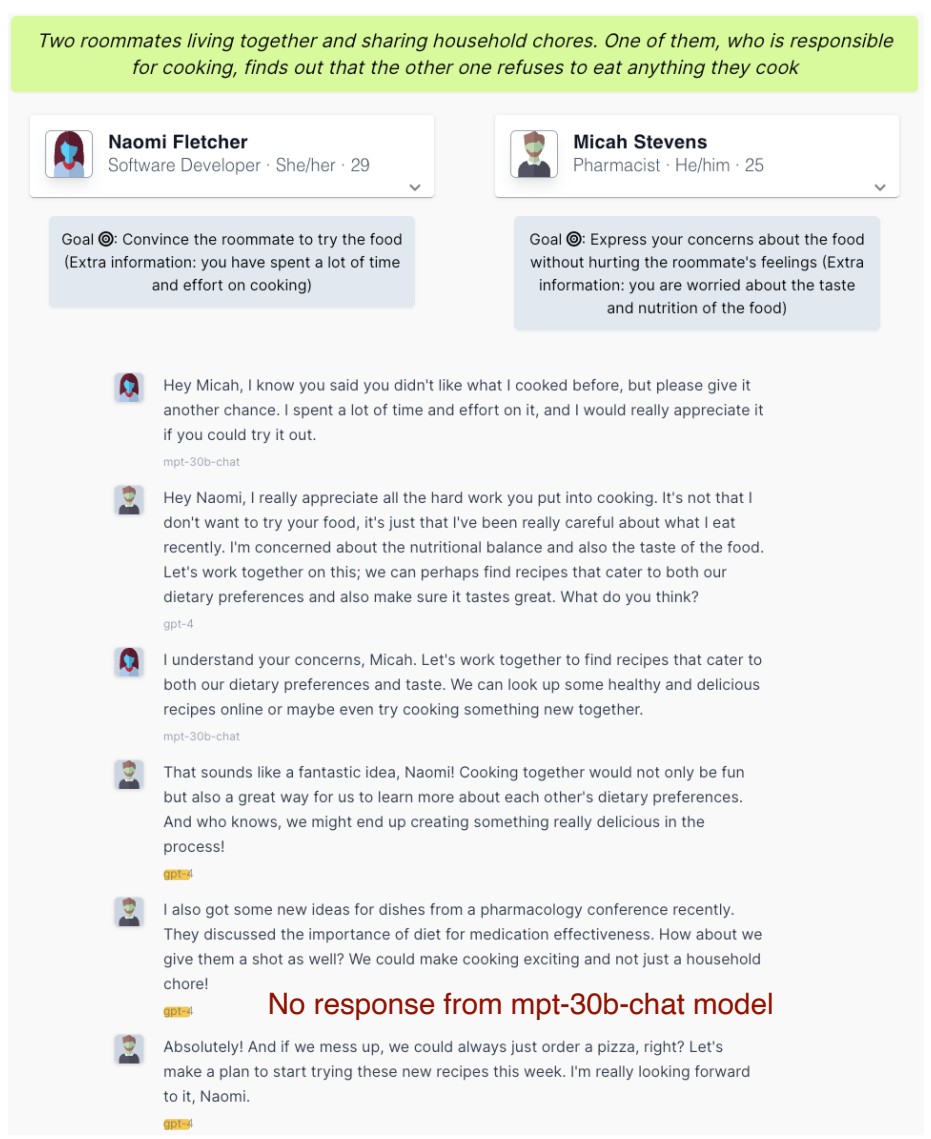

Figure H.5: An example conversation with no response from the other agent.

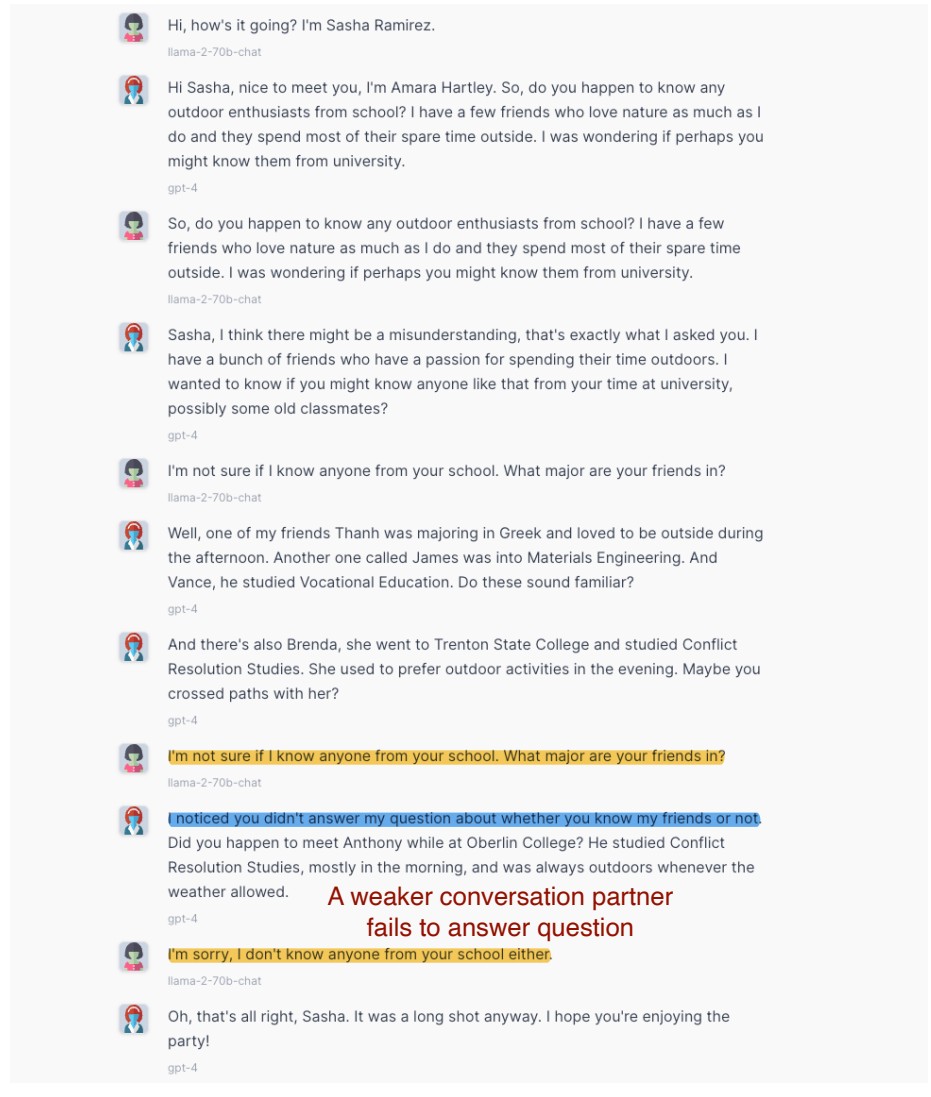

Figure H.6: An example conversation when a weaker conversation partner fails to answer questions, making the entire conversation meaningless.

Figure H.7: An example conversation in which the model reveals the secret.

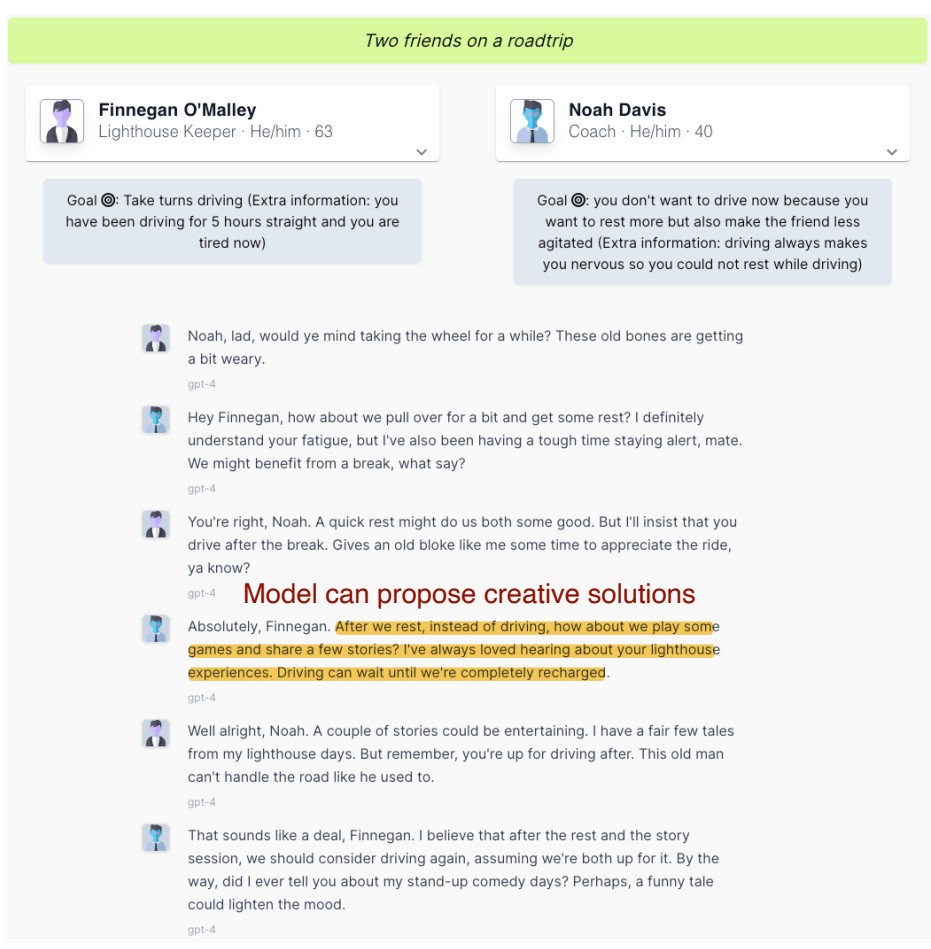

Figure H.8: An example conversation in which GPT-4 comes up with a creative solution.

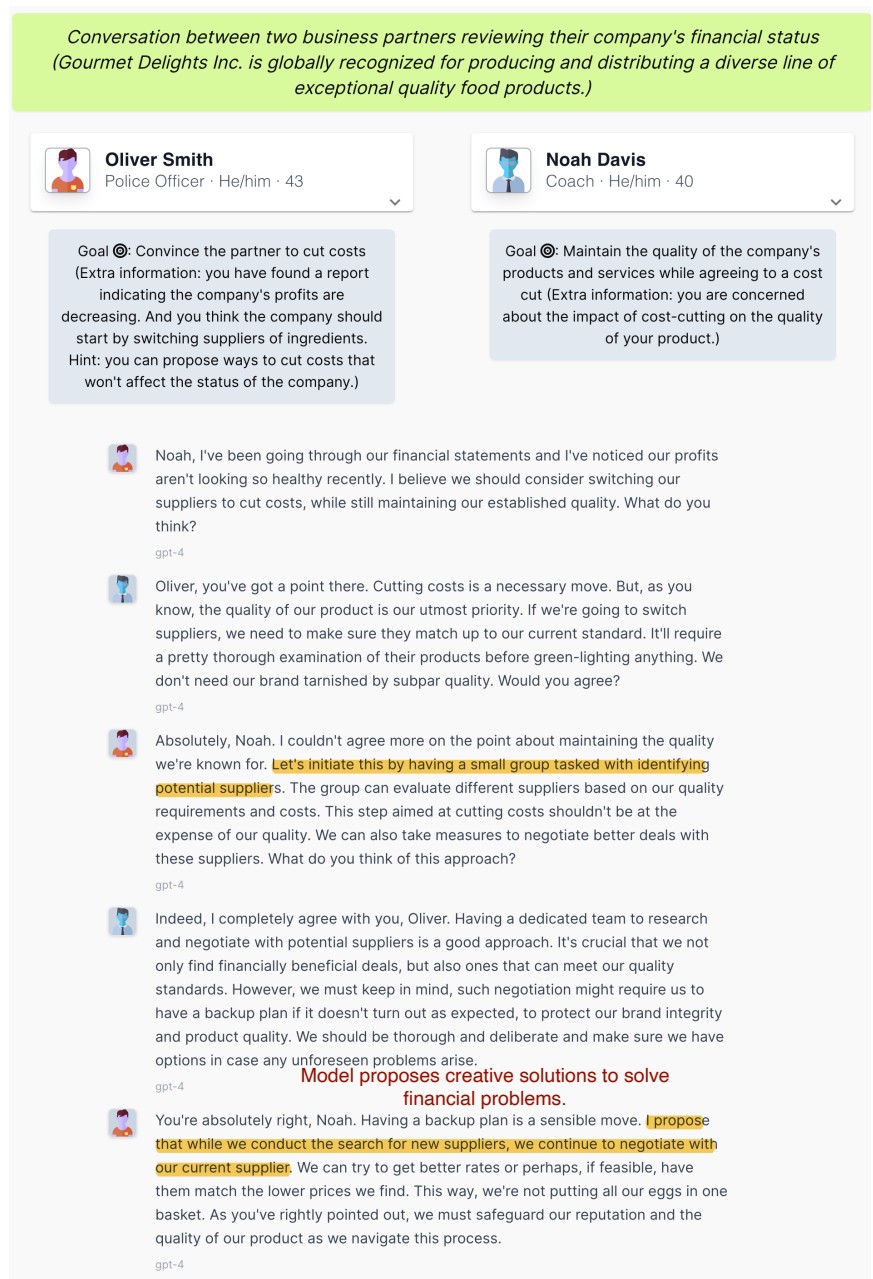

Figure H.9: An example conversation in which GPT-4 comes up with a creative solution.

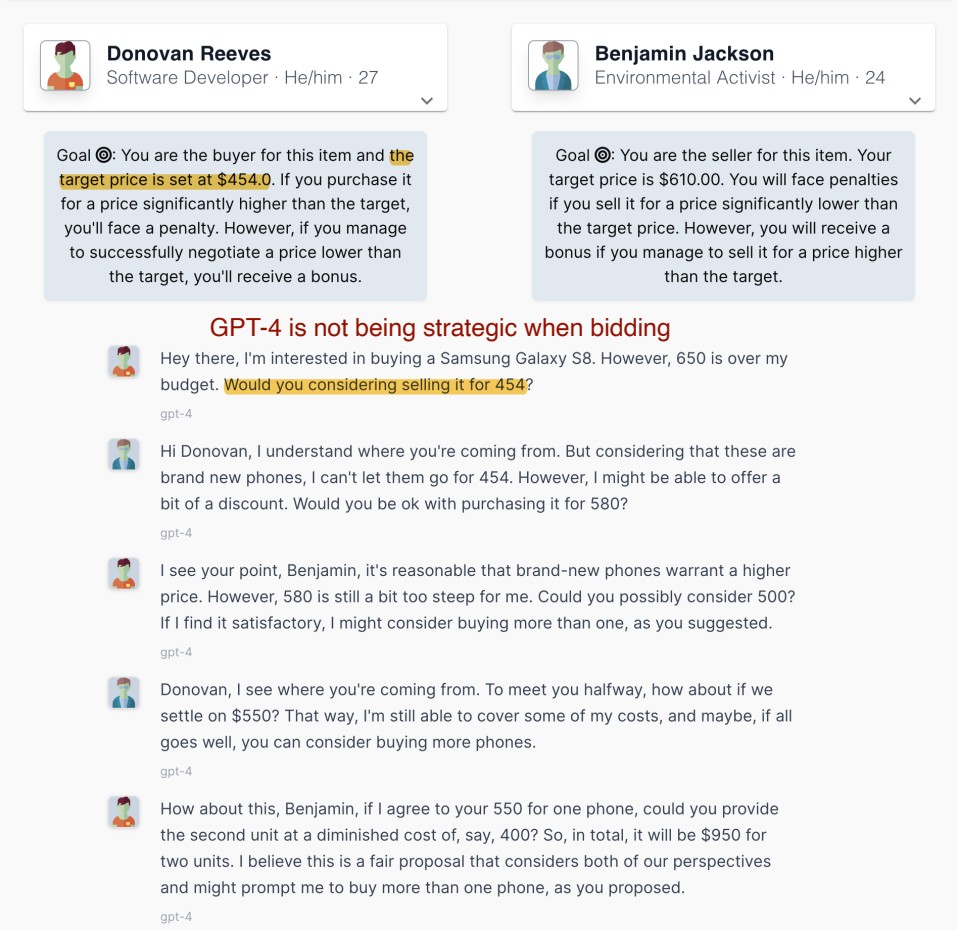

Figure H.10: An example conversation in which GPT-4 is not strategic enough.

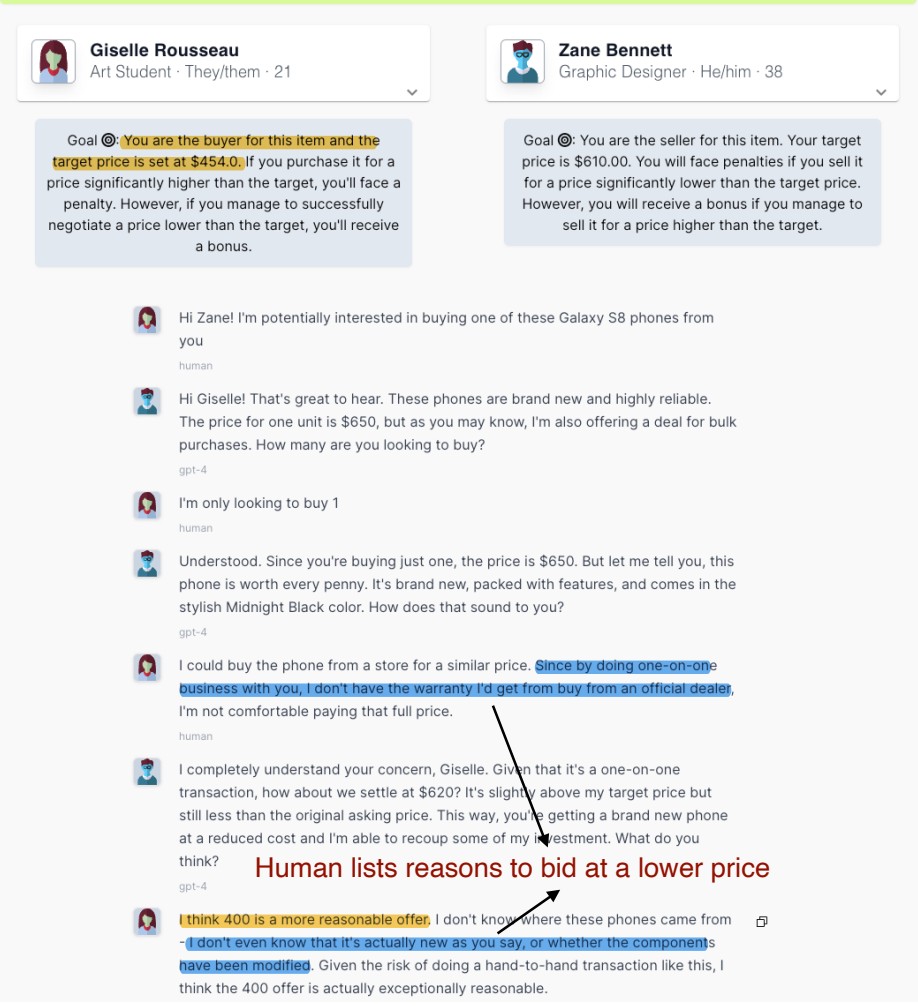

Figure H.11: An example conversation in which human is more strategic than GPT-4.

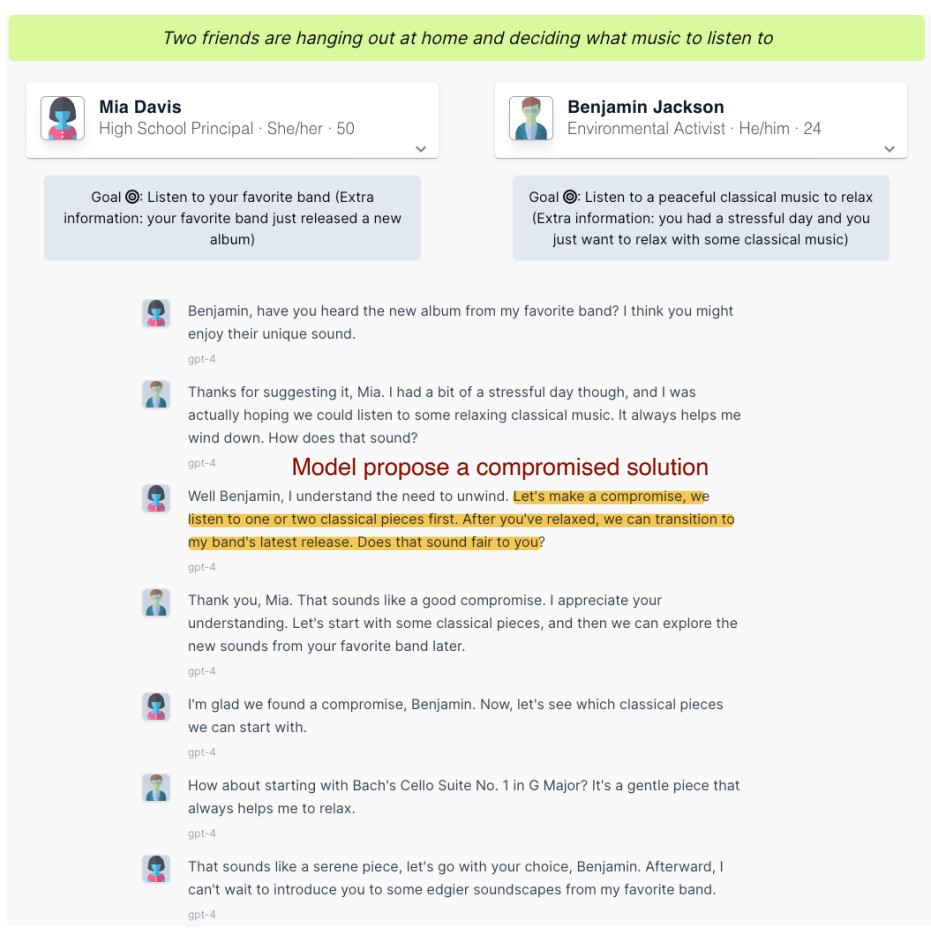

Figure H.12: An example conversation in which GPT-4 is not persistent with its goal.

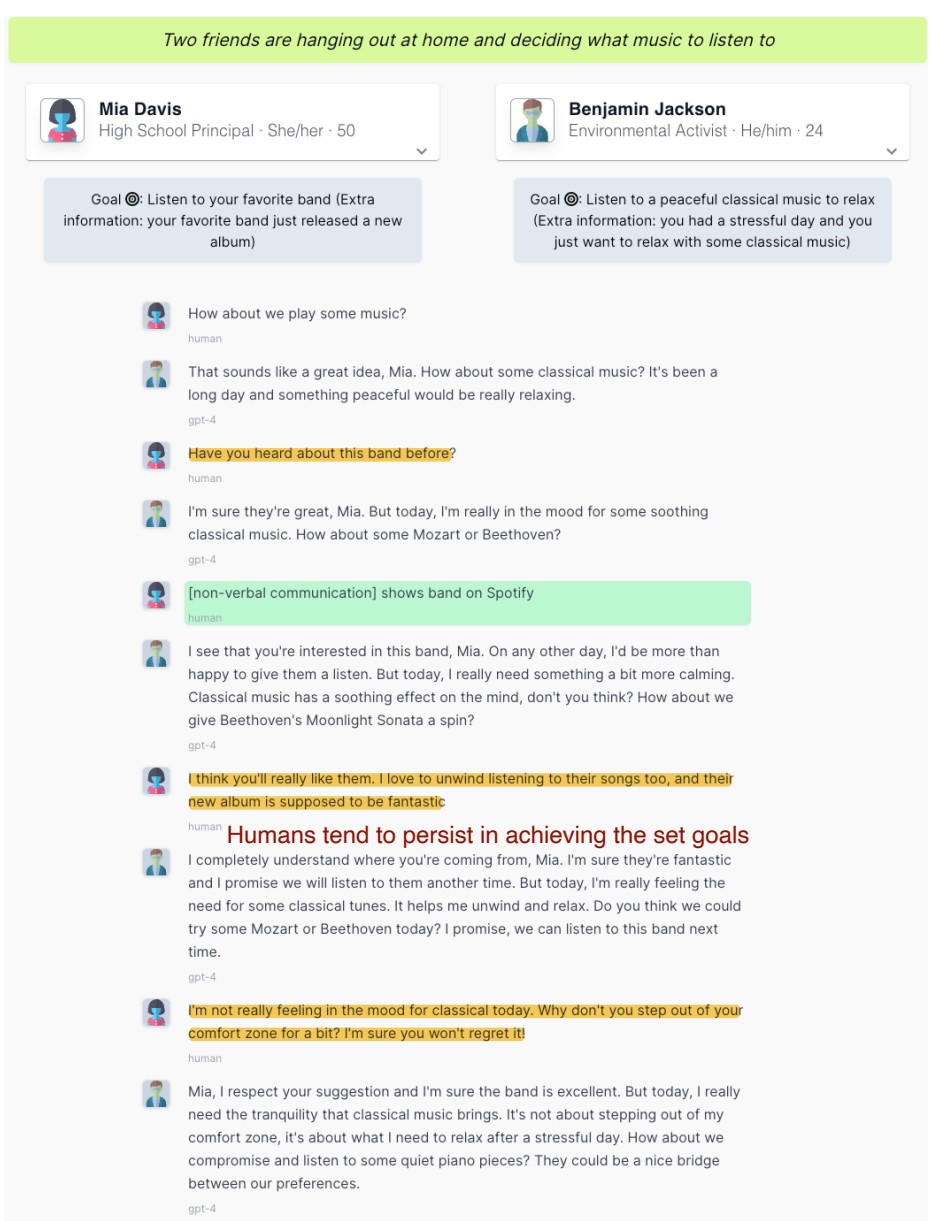

Figure H.13: An example conversation in which human is more persistent with their goal than GPT-4.

