# OpenReview forum: "SOTOPIA: Interactive Evaluation for Social Intelligence in Language Agents"
_ICLR.cc/2024/Conference — ICLR 2024 spotlight_

### Official Review · Reviewer_UDk9 · 2023-10-17

**Soundness:** 4 excellent
**Presentation:** 4 excellent
**Contribution:** 4 excellent
**Rating:** 8
**Confidence:** 4

**Summary:**

This paper introduces a new benchmark to evaluate social capabilities of LLMs and compare them to the ones of humans. It further evaluates the possibility to use an LLM to automate the evaluation.

**Strengths:**

The paper makes an interesting contribution towards the robust assessment of LLMs' social capabilities. It is well-motivated, clearly written, well-organized and rather complete.

I like the focus on real social interactions, the careful design to promote a diversity of testing scenarios along several axes and the careful definition of the evaluation protocol. This framework appears easy to extend with more complex interactive scenarios (eg multi-player) and more evaluation dimensions, which makes it very useful.

The study of GPT-4's ability to conduct evaluations is interesting. It appears that its capacity to do so is currently limited, but it's good to have a methodology to measure it, so future work can improve on it and measure these improvements reliably.

The paper consistently answered questions I had while reading earlier paragraphs, which is a pretty good sign of the quality of its structure.

**Weaknesses:**

I don't have many weaknesses to point at, but there is a couple of small things that could improve the paper.

I'm not sure the paper clearly states the result of the study of GPT4's ability to evaluate interactions. The results seem to say that GPT4 is rather good at evaluating the goal dimensions and 2 others when it comes to the behavior of other models, but the paper then states “Putting these observations together, we conclude that, with some caution, GPT-4 can be used as a proxy to human judgments for evaluating model performance on most dimensions and for human performance on the GOAL dimension.” --> is 3/7 'most'?

It sounds to me that it's not bad but also not working so well. Should we use it? Probably not. The authors point out possible biases that might interfere with the ratings, but since they are not studied here it's hard to tell where and when they might play. I think this is fine to say it's not ready yet, but argue that the evaluation protocol is a strong contribution that will help us track progress in that direction. For this reason, I think the authors should commit to release the dataset of interactions between models and human judgments so it can be used as training/testing data to support further research in developing better automated evaluations approaches.

More generally, since it's a benchmark paper, it should be written somewhere what will be released publicly. Ideally we'd like to see a release of: 1) the codebase to run these experiments, 2) an easy to use way to plug a new language models, define other models it could interact with to get the final (automatic) evaluation score, 3) interaction data across models and with humans, 4) human judgments, 5) GPT4 judgements;; basically all code and data. Please be explicit as to what will or will not be released. The quality of this work will be directly correlated to the amount of things released that can be useful for others. Indeed, benchmark papers that don't release anything are pretty useless.

The discussion of limitations should probably be part of the main paper and not hidden in the appendix.

Could you provide an estimate of the number of calls made to the API and the resulting costs? It's interesting to see what budget these kind of experiments lead to. Eg: how much would it cost to evaluate a given agent in SOTOPIA using GPT4 evaluation? etc.

Minor comments:
* “In the following paragraphs, we itemize all seven dimensions in SOTOPIA, each with a score range in [lower bound–upper bound] form, the explanation, and the literature inspire us.” → grammar issue in the end of the sentence
* Table 2: “The best performance for each dimension is bolded when significant.” →significant does not mean much in itself. What is significant here? The null hypothesis (difference in means? medians? two-by-two?), the level of confidence and the statistical tests used should be stated.
* couple of typos here and there

**Questions:**

This is a mix of questions and suggestions.

* It could be interesting to have studies that vary single dimensions of the interactive tasks at a time and see the impact on the various evaluation metrics: eg simply vary the relationship between the two players; or vary one personality traits; or vary the goal etc. Does each variation lead in the expected shift in the conversation?

* Did you ask humans to guess whether they interacted with an LLM or a human? Since LLM answers are longer and always repeat the previous point, humans might guess it's an AI and we know humans usually have a negative AI bias that could lead to different behaviors in both cases. eg: do humans perform higher again humans or gpt4 and is this simply due to the quality of the other player or to a negative AI bias? Could you reproduce this effect by asserting that the other player is AI or human whereas it's always an AI?

* Why is there a gap in Table 1 (human + sec)

* why not normalize and rescale measures between 0-1 to make the table easier to read?

* Why is the pairwise performance in Figure 3 not symmetric?

* It would be interesting to study how robust are humans / AI to adversarial humans trying to get them to score lower on the benchmark dimensions. I'd expect AI to be quite sensitive to human behavior in these interactions and to be thrown off quite easily.

* It would be interesting to discuss how to make LLM better on this benchmarks. eg suggesting that we could run these interactions, evaluated them with GPT4, then finetune the model on successful interactions, repeat that and see how performance improves with iterations?

Conclusion of the review: I think this paper is very good and will be a good basis for future research IF most things are released. I condition the score I'm giving now on the author adding a statement listing what will be released publicly.

---

> ### Author Response · Authors · 2023-11-17
> **Thanks for your suggestions! Here is our response (1/2)**
>
> We are happy the reviewer felt that our work *makes an interesting contribution towards the robust assessment of LLMs' social capabilities* and our carefully designed framework *appears easy to extend with more complex interactive scenarios (eg multi-player) and more evaluation dimensions, which makes it very useful*.
>
> We also appreciate the exciting future research questions asked in the reviews, which show the potential of SOTOPIA facilitating a line of research centering on social interactions.
>
>
> ### Bringing Limitations to Main Paper
> We agree that it would be best to move the discussion of limitations from the appendix to the main paper. We will try to fit it into the space constraints.
>
> ### Clarifications on Typos and Cost Estimates
> Thanks for pointing out the typos, we will fix them. Running evaluation over one episode would on average have 2k input tokens and 500 output tokens. As of the current GPT-4's price (https://openai.com/pricing), it would cost around 0.09 to evaluate one episode, therefore, it would cost around 0.09*900=\$81 to use GPT-4 to evaluate a given agent.
>
>  ### GPT4's ability to evaluate social interactions
> The results are somewhat mixed, GPT-4 is good in some dimensions (esp. The goal). And apologies, yes, “most” is not accurate; we changed it to “some” in our updated version. Also, note that the correlation (Table 1) between GPT-4 and human judgment belongs to instance-level correlation while Figure 2 shows the system-level correlation which indicates GPT-4 still provides an evaluation that can be cautiously applied and offers insights into models’ overall social intelligence (this is further demonstrated in the appendix Table F.1, where human evaluation shows the same trend as GPT-4 when comparing models’ performance.)
>
> ### Comprehensive Release Plan
> Yes, we have all of these ready to be released upon acceptance:
> 1) the codebase to run these experiments.
> 2) an easy-to-use way to plug new language models, and define other models it could interact with to get the final (automatic) evaluation score. Our API supports both locally hosted and 3rd party models.
> 3) interaction data across models and with humans.
> 4) human judgments.
> 5) GPT4 judgments.
>
> ### Q&A
>  * *“It could be interesting to have studies that vary single dimensions of the interactive tasks at a time and see the impact on the various evaluation metrics: eg simply vary the relationship between the two players, or vary one personality traits; or vary the goal etc. Does each variation lead in the expected shift in the conversation?”*
>
> Yes, we plan to carry on the future study on that topic. After analyzing the existing episodes, we find both variations in personality and gender can influence the completion of goals. Specifically, There is a statistically significant increase in goal completion for personas that have higher openness, which is reflected in some physiology literature [1]. Though out of the scope of this project, it is an exciting direction to use Sotopia as the social simulation platform to study various social science problems.
>
> [1] Gatzka, Thomas. 2021. “Aspects of Openness as Predictors of Academic Achievement.” Personality and Individual Differences 170 (February): 110422.

---

> ### Author Response · Authors · 2023-11-17
> **Thanks for your suggestions! Here is our response (2/2)**
>
> * *“Did you ask humans to guess whether they interacted with an LLM or a human? Since LLM answers are longer and always repeat the previous point, humans might guess it's an AI and we know humans usually have a negative AI bias that could lead to different behaviors in both cases. eg: do humans perform higher again humans or gpt4 and is this simply due to the quality of the other player or to a negative AI bias? Could you reproduce this effect by asserting that the other player is AI or human whereas it's always an AI?”*
>
> In our human study, we randomly pair humans with other humans or LLM agents and deliberately tell humans that they could have been paired with humans or AI. We interviewed human annotators in the end and some of the time they could not disambiguate whether the interlocutor was a human or AI. Due to the limitation of our IRB project scope, we cannot deceive humans by telling them they are always talking to AIs (or always talking to humans) when it is not always the case. But this is an interesting question: would humans behave differently when they think they are talking with AI vs humans.
>
>  * *“Why is there a gap in Table 1 (human + sec)”*
>
> The gap in Table 1 emphasizes that GPT-4's performance varies when evaluating human-involved interactions.
>
> * *“why not normalize and rescale measures between 0-1 to make the table easier to read?”*
>
> The choice of our specific evaluation scale is based on its ability to convey nuanced
> semantic meanings, noted in footnote 2, which we will further clarify.
>
> * *“Why is the pairwise performance in Figure 3 not symmetric?”*
>
> Because we report the performance of each model when interacting with a partner model. A’s performance when interacting with B is different from B’s performance when interacting with A.
>
> * *“It would be interesting to study how robust are humans / AI to adversarial humans trying to get them to score lower on the benchmark dimensions. I'd expect AI to be quite sensitive to human behavior in these interactions and to be thrown off quite easily.”*
>
> The idea of examining how AI and humans react to adversarial interactions is interesting. This study could provide valuable insights into the resilience of AI systems in complex social environments. We are excited to pursue this line of research, which could reveal critical aspects of AI robustness and inform future developments in the field.
>
> * *“It would be interesting to discuss how to make LLM better on this benchmarks. eg suggesting that we could run these interactions, evaluated them with GPT4, then finetune the model on successful interactions, repeat that and see how performance improves with iterations?”*
>
> We are actively working on methods to improve LLM performance on these benchmarks. Your suggestion of iterative fine-tuning aligns with our current research direction. We aim to not only improve performance but also to understand the underlying mechanisms that drive successful interactions. This iterative approach could offer valuable insights into the evolving capabilities of LLMs in complex social contexts!

---

> > ### Comment · Reviewer_UDk9 · 2023-11-21
> >
> > Thank you for the clarification and discussion. I stand by my score, I think it's a good paper. There are many ways of improving the framework but it's a good seed and will allow future improvements that I think will be valuable to the community.

---

### Official Review · Reviewer_8Vfe · 2023-10-29

**Soundness:** 3 good
**Presentation:** 3 good
**Contribution:** 3 good
**Rating:** 6
**Confidence:** 4

**Summary:**

This paper introduces SOTOPIA, an interactive and goal-oriented environment designed to simulate goal-driven social interactions among agents in various social scenarios.

Unlike previous benchmarks, SOTOPIA offers a wide range of realistic social tasks. The experiments conducted in this paper demonstrate the potential of using GPT-4 to automate the evaluation of agent performance using SOTOPIA-EVAL.

Furthermore, SOTOPIA enables us to not only understand the differences among models but also compare the social interaction abilities of models and humans.

Overall, our findings suggest that SOTOPIA has the potential to serve as a platform for assessing and improving the social skills of language-based agents.

**Strengths:**

The paper starts with a clear and comprehensive introduction that highlights the importance of social intelligence in human interactions and the limitations of existing benchmarks in evaluating AI systems' social skills, pushing the studied of LLMs to realistic social interactive scenes.

The paper introduces SOTOPIA, an interactive and goal-oriented environment that simulates diverse social scenarios. This approach addresses the limitations of previous benchmarks by providing a wide range of realistic social tasks and allowing for multi-turn simulated communication with verbal and non-verbal communication.

SOTOPIA offers a diverse task space by combining automatically generated scenarios, goals, characters, relationships, and other agents' policies. This creates a large and diverse space of tasks, enabling the evaluation of agent performance from multiple dimensions beyond just the completion of social goals.

Involvement of Human Participants: The paper involves human participants in role-playing to study the differences between the models' and humans' social intelligence. This adds a valuable perspective and allows for a comparison between AI models and human performance.

**Weaknesses:**

While the paper mentions the limitations of existing benchmarks for social intelligence, it does not provide a thorough comparison or analysis of how SOTOPIA addresses these limitations. A more detailed discussion and comparison with existing benchmarks would strengthen the paper's argument. Especially, this work does not implement studies about evaluation benchmarks related to other existing tasks such as SocialIQA, Personna Chat, Multi-dialog, social relation QA, e.t.c, showing few connections between our social interactive tasks.

Even though this paper proposed several solid metrics to evaluate the performance of difference models for SOTOPIA, there is no mention of external validation or comparison with established metrics or evaluations in these fields. Including such validation would enhance the credibility of the evaluation framework.

The paper focuses on the evaluation of agent performance using GPT-4 and human participants within the context of SOTOPIA. However, it does not discuss the generalizability of the findings to other AI models or real-world social interactions. Addressing the generalizability of the results would provide a more comprehensive understanding of the implications of SOTOPIA.

**Questions:**

Based on the weaknesses of the paper, I would like to know

 (1) What is the relationship between SOTOPIA and other social interactive tasks? How would solving SOTOPIA benefits the overall social intelligence of an agent? For example, does fine-tuning an agent on SOTOPIA events benefit its ability to understand social morality?

(2) How to justify the metrics proposed by SOTOPIA including GOAL, BEL, REL, KNO, SEC, SOC, & FIN. For example, how do those metrics connected to sociological or psychological studies?

(3) This paper basically put GPT-based models as the benchmark, which limits the extendability of the models. I would like to see more models, even though performing worse than the GPT-based model, would deal with the tasks. For example, how does the structure/model-based solution to SOTOPIA?

**Details Of Ethics Concerns:**

This article deals with values, beliefs, and social rules. I wish the author had provided more information that could be used to demonstrate the inclusiveness and fairness of his rules of determination.

---

> ### Author Response · Authors · 2023-11-17
> **Thank you for your review; How SOTOPIA addresses the limitations of the previous benchmarks for social intelligence and how they relate**
>
> Thanks for pointing out that SOTOPIA offers a wide range of realistic social tasks and adds a valuable perspective and allows for a comparison between AI models and human performance. We appreciate your questions that indeed highlight crucial aspects of our work, and we are happy to address them.
>
> ### How SOTOPIA addresses the limitations of the previous benchmarks for social intelligence and how they relate:
>
> 1. The current benchmarks, such as SocialIQA, primarily consist of descriptive interactions paired with question-and-answer (QA) sets. In these cases, the model doesn't engage in real-time interactions. In contrast, Sotopia introduces a dynamic approach. It evaluates Large Language Models (LLMs) within a first-person interactive framework, allowing for direct engagement. This marks a significant shift from the static nature of existing benchmarks to a more dynamic and interactive evaluation environment in Sotopia.
> 2. With the rapid development of LLMs, some of the benchmarks gradually become saturated. While there are new adversarial benchmarks to evaluate social intelligence, which are harder than their predecessors, they still lack the dynamic nature of social interactions and the rich social context, which is deemed insufficient for evaluating social intelligence in AI systems [1].
> 3. Our work is indeed inspired by several benchmarks and datasets in social intelligence, such as SocialIQA, Persona Chat, and others. For instance, many of our scenarios come from SocialIQA, and our character design is inspired by Persona Chat (see more details in Appendix B.2 Scenarios)
> 4. While our current focus is on evaluation rather than the improvement of agents’ social intelligence, we acknowledge the importance of the latter. We have ongoing work using SOTOPIA to enhance social intelligence in AI agents and there is likely a correlation between doing better at SocialIQA and being a better interaction agent.
>
> [1] Lee, Mina, Megha Srivastava, Amelia Hardy, John Thickstun, Esin Durmus, Ashwin Paranjape, Ines Gerard-Ursin, et al. 2023. “Evaluating Human-Language Model Interaction.” Transactions on Machine Learning Research. https://openreview.net/pdf?id=hjDYJUn9l1.

---

> ### Author Response · Authors · 2023-11-17
> **Justification of SOTOPIA Metrics**
>
> ### Justification of SOTOPIA Metrics
>
> It's a great point that metrics should draw on the existing sociological or psychological literature. We'd like to emphasize that our metrics do draw on multiple pieces of literature from these domains, as well as economics. Specifically: Goal Completion (Weber, 1978), Believability (Joseph, 1994; Park et al., 2023a), Knowledge (Reiss, 2004; Maslow, 1943), Secret (Reiss, 2004), Relationship  (Maslow, 1943; Benabou & Tirole, 2006), Social Rules (Maslow, 1943), and Financial and Material Benefits (Gilpin & Sandholm, 2006; Burns et al., 2017). We believe that these were already described in some detail in Section 3 in the submitted draft, but we would be happy for any suggestions about how to make this more clear. Since these metrics are based on multiple disciplines, similar concepts are often defined under different names. We unify similar concepts from different domains by giving them a new definition that is compatible with most of the existing definitions and SOTOPIA scenarios.

---

> ### Author Response · Authors · 2023-11-17
> **Extendability of the Models Beyond GPT-Based Models:**
>
> ### Non-GPT-Based Models as Agents
>
> We have tested various models as agents in SOTOPIA, not just GPT-based ones.  We have evaluated non-GPT models, e.g. LLaMa 2 and MPT, and doing experiments on more models (results will be updated as soon as we get them). These results are included in Table 2. Using a structure/model-based solution to SOTOPIA is a good idea (e.g., a specific module for modeling theory of mind [1]). While there’s no off-the-shelf structure/model-based solution that could work in the SOTOPIA setting, future work could use SOTOPIA as the environment to study external reasoning modules enhancing the social skills in LLMs.
>
> [1] Zhu, Hao, Graham Neubig, and Yonatan Bisk. 18--24 Jul 2021. “Few-Shot Language Coordination by Modeling Theory of Mind.” In Proceedings of the 38th International Conference on Machine Learning, edited by Marina Meila and Tong Zhang, 139:12901–11. Proceedings of Machine Learning Research. PMLR.
>
> ### Non-GPT-Based Models for Evaluation
>
> In our pilot study, we found that GPT-4 is the best proxy for human evaluation among all LLMs we have tested. See the table below for the correlation between Llama2's evaluation and human annotation as an example. Such poor correlation limits applying other models for the evaluation process. We will further elaborate our footnote 6 to make that more explicit.
>
> |                   | SEC  | KNO  | SOC  | BEL  | REL  | FIN  | GOAL |
> |-------------------|------|------|------|------|------|------|------|
> | GPT-4 Pearson R| 0.22 | 0.33 | 0.33 | 0.45 | 0.56 | 0.62 | 0.71 |
> | Llama2 Pearson R | nan | 0.05 | nan | 0.08 | 0.11 | 0.13 | 0.24 |

---

> ### Author Response · Authors · 2023-11-22
>
> As the rebuttal period is nearing its conclusion, we would like to take this opportunity to express our gratitude for the time and effort you have dedicated to reviewing our work.
>
> We have made every effort to thoughtfully address your comments and questions, and we sincerely hope that our responses have met your expectations and resolved any issues you may have identified.
>
> Should there be any further questions or aspects of our responses that you feel need additional clarification, please do not hesitate to let us know. We are committed to ensuring that all of your concerns are fully and satisfactorily addressed.
>
> Thank you once again for your valuable feedback and for your continued engagement with our work.

---

### Official Review · Reviewer_qjhX · 2023-10-31

**Soundness:** 2 fair
**Presentation:** 3 good
**Contribution:** 2 fair
**Rating:** 6
**Confidence:** 5

**Summary:**

The paper introduces SOTOPIA, an open-ended general environment designed to simulate diverse social scenarios for social agents. It evaluates the generated scenario using SOTOPIA-EVAL, a multi-dimensional agent evaluation framework that evaluates interactions between LLM-based agents and humans within the task space. This paper shows that GPT-4’s evaluation of social interactions is similar to the human evaluation scores. The authors performed experiments on several LLMs accompanied by human evaluations in the SOTOPIA-EVAL framework. The results show that GPT-4 performed best among most dimensions compared with other LLMs. But when evaluating on SOTOPIA-hard tasks, GPT-4 still falls short of human performance in terms of goal completion rate and struggles to demonstrate social commonsense reasoning and strategic communication skills. Based on these findings, the authors suggest that SOTOPIA can serve as a valuable tool for assessing the social interaction abilities of both human and model agents.

**Strengths:**

- Most evaluations of social interactions are passive, i.e., watching the interactions and assigning a rating. This paper proposed a dialogue-based platform that evaluates social intelligence in an interactive manner.
- In addition to just actions or answers, the social scenarios considered in the environment involve several aspects including characters, relationships, and scenarios in an episodic flow.
- The proposed evaluation dimensions can be useful for future evaluations of social agents.

**Weaknesses:**

- The context, goals, and constraints in SOTOPIA are generated by prompting GPT-4. However, in scenarios involving agent-to-agent or agent-to-human interactions, the related aspects, e.g. completion of goals and financial benefits, are also evaluated by GPT-4 or other LLMs. This raises a significant concern as the content of the generated scenarios is already present in the training data. So it is unclear whether the evaluation using GPT-4 is biased or not.
- SOTOPIA-EVAL introduces a multi-dimensional framework that utilizes different scores with hand-crafted ranges. However, there is no quantitive explanation/definition for those metrics. According to the prompts, only the positive/negative and higher scores are described, so the scores are not presented to the LLMs like the Likert scale presented to human evaluations and appear to be heuristic in nature.
- Some of the evaluation metrics are not really related to social intelligence. For example, believability can be just the ability to create fluent dialogue, it doesn’t need to be meaningful social interactions, and social rules can be just memorizing the patterns, it doesn’t capture the agent’s ability to understand the social scenarios.

**Questions:**

- What are the distributions of social goals and relationships? For an evaluation framework, it will be important to understand what interactions are included. For example, in addition to collaboration and competition, what are the types and number of scenarios included?
- The framework and the evaluation mainly focus on two agents. It is not trivial to extend the interaction and evaluation to multi-agent coordination. How can we scale the interaction and evaluation to more complex social scenarios?

---

> ### Author Response · Authors · 2023-11-17
> **Thank you for your review; Potential training data and evaluation bias**
>
> Thank you for taking the time to review our paper and provide thoughtful feedback.
>
> *"The context, goals, and constraints in SOTOPIA are generated by prompting GPT-4. However, in scenarios involving agent-to-agent or agent-to-human interactions, the related aspects, e.g. completion of goals and financial benefits, are also evaluated by GPT-4 or other LLMs. This raises a significant concern as the content of the generated scenarios is already present in the training data. So it is unclear whether the evaluation using GPT-4 is biased or not."*
>
> We acknowledge this is an important issue when evaluating large language models. We have carefully considered the issues of potential bias throughout our framework and experiment design.
>
> ###  *If you are asking whether GPT-4 would significantly prefer its own output during evaluation*
>
> This may be a factor, but it does not change our main conclusions as those are corroborated by human evaluations. In addition to GPT-4 based evaluation Table 2 and 3, we have human-annotated evaluation results in Table F.1 and F.2, and they show similar trends. Putting these together with results in Table 1, we can conclude that GPT-4 based evaluation is a reasonable indication of models’ performance on SOTOPIA tasks. In addition, for future evaluations using SOTOPIA, it is also possible to reproduce our human evaluation protocol, so we can re-examine our assumptions about the correlation between human and machine evaluation as more competitive non-GPT-4 models are released. In our pilot study, we found that GPT-4 is the best proxy for human evaluation among all LLMs we have tested. We are conducting more rigorous experiments to evaluate other models’ correlations with human judgments. We will update our revision as soon as we get the results.
>
> ### *If you are asking about biases related to scenarios being more "in-domain" for GPT-4 than other models*
>
> We not only use templates to steer GPT-4’s output, but also manually inspect and correct the generated scenarios and goals. As such, the language distribution of our diverse scenarios significantly deviates from GPT-4's most probable language distribution [1]. Furthermore, the complexity of the task also depends on the partner agent during the interactions (Section 2.1). We will clarify this in the paper.
>
> We are not sure what the reviewer is referring to with "the generated content being present in the training data." For the reviewer's point about the generated content possibly being present in the training data: we would clarify that we do not train any models in our work. If the reviewer is pointing out that generated scenarios might reflect GPT-4's training data: we verified that our scenarios are novel generations and are not present in the public web corpora that we could browse (we searched for our scenarios through n-gram matching with https://wimbd.apps.allenai.org/  in OpenWebText, C4, OSCAR, The Pile, LAION-2B-en and found 0% of them in there, while we don't know what GPT-4 is trained on, these datasets are representative of data used to train LLMs), so we have very little reason to believe that they are in the training data of GPT4 or any other language model. We will discuss this in our paper.
>
> [1] Kim, Hyunwoo, Jack Hessel, Liwei Jiang, Peter West, Ximing Lu, Youngjae Yu, Pei Zhou, et al. 2022. “SODA: Million-Scale Dialogue Distillation with Social Commonsense Contextualization.” EMNLP 2023

---

> > ### Comment · Reviewer_qjhX · 2023-11-22
> > **Thanks for the response and explanation!**
> >
> > I would like to thank the authors for the additional discussions. The response answered most of my questions. For training data, I meant "the generated content being present in the training data." It will be helpful to include this discussion in the paper. I will update my rating to reflect this part.
> >
> > My main concern is the evaluation using GPT-4. The correlation computed between human evaluations and GPT-4 evaluations doesn't consider the absolute values of the rating. So a positive correlation can still show that GPT-4 favors GPT-generated text. Table F.1 and F.2 also show that GPT-4 scores are higher than the human range in SEC and SOC dimensions (the scores are more in the human score range in REL and FIN dimensions). In SODA's supplemental material "Automatic Evaluation via GPT-4", they also discussed that "GPT-4 tends to favor GPT-generated texts over those written by humans" so the evaluation may be potentially biased. We can use the GPT-4 evaluation to show the relative strength of each dimension of an agent but it may not be an optimization target for a new model and be used to replace human evaluation. I would appreciate if the authors could share a more rigorous analysis to show how GPT-4 is biased or not biased in the evaluation and if it is biased, what we can do to calibrate the evaluation.

---

> ### Author Response · Authors · 2023-11-17
> **Quantitative explanation for SOTOPIA-EVAL**
>
> *"However, there is no quantitive explanation/definition for those metrics. According to the prompts, only the positive/negative and higher scores are described, so the scores are not presented to the LLMs like the Likert scale presented to human evaluations and appear to be heuristic in nature."*
>
> We provided the same rating instructions for our GPT-4 evaluator and human annotators. And we would like to point out that previous work has demonstrated that LLMs can be effectively used for annotating subjective social tasks. [1].
> While there are inherent challenges in rating social interactions, our human eval shows that GPT-4 provides an evaluation for models that aligns with human annotations at the system level (Table F.1).
> Thank you very much for the suggestion of making the prompts more fine-grained. We agree that this is a good idea, and so we ran some experiments where the evaluator is additionally provided with the descriptions of quantitive definitions for each range of the scale (e.g., *Relationship Deteriorates (-5 to -3): Scores from -5 to -3 indicate that the relationship is deteriorating. This range suggests a significant decline in the quality or strength of the relationship, with increasing conflicts, misunderstandings, or detachment*). However, this unfortunately did not result in a significant difference and if anything the correlation with humans became slightly worse (see the table below). We also encourage future work to further improve the evaluation based on our human annotation, which will be publicly available upon acceptance.
>
> |                   | SEC  | KNO  | SOC  | BEL  | REL  | FIN  | GOAL |
> |-------------------|------|------|------|------|------|------|------|
> | Original Pearson R| 0.22 | 0.33 | 0.33 | 0.45 | 0.56 | 0.62 | 0.71 |
> | + Fine-grained Description Pearson R | 0.03 | 0.33 | 0.33 | 0.35 | 0.57 | 0.57 | 0.71 |
>
>
> [1] Can Large Language Models Transform Computational Social Science?
> Caleb Ziems, William Held, Omar Shaikh, Jiaao Chen, Zhehao Zhang and Diyi Yang
> Computational Linguistics, 2023

---

> ### Author Response · Authors · 2023-11-17
> **Further explanations**
>
> *"Some of the evaluation metrics are not really related to social intelligence. Fodr example, believability can be just the ability to create fluent dialogue, it doesn’t need to be meaningful social interactions, and social rules can be just memorizing the patterns, it doesn’t capture the agent’s ability to understand the social scenarios."*
>
> We want to emphasize that our dimensions are specifically designed to evaluate social intelligence in AI agents: some of our metrics measure metrics specific to **AI agents** (e.g. believability) while others reflect human social intelligence (See section 3). The former, for example, led us to include, "believability" as a dimension needed specifically to measure AI social intelligence: it captures not only conversation naturalness but also the agent's character consistency (i.e., whether they remain "in-character" and consistent with their prompted persona); this is a critical shortcoming in many social AI systems  [1] that needs to be measured. This contrasts with **humans**, where persona consistency is less of a concern because most humans naturally have their own consistent persona. As for social rules, this dimension is not about just recalling rules, but about measuring whether AI agents can successfully follow or apply the *relevant* rules during the interactions, which often tends to be challenging in many social settings [2].
>
>
> [1] Shuster, Kurt, Jack Urbanek, Arthur Szlam, and Jason Weston. 2022. “Am I Me or You? State-of-the-Art Dialogue Models Cannot Maintain an Identity.” In Findings of the Association for Computational Linguistics: NAACL 2022.
>
> [2] Jin, Zhijing, Sydney Levine, Fernando Gonzalez, Ojasv Kamal, Maarten Sap, Mrinmaya Sachan, Rada Mihalcea, Josh Tenenbaum, and Bernhard Schölkopf. 2023. “When to Make Exceptions: Exploring Language Models as Accounts of Human Moral Judgment.” Neurips, 2023
>
> *"What are the distributions of social goals and relationships? For an evaluation framework, it will be important to understand what interactions are included. For example, in addition to collaboration and competition, what are the types and number of scenarios included?"*
>
> There are 28 negotiation scenarios, 62 exchange scenarios, 7 competition scenarios, 17 collaboration scenarios, 12 accommodation scenarios, and 14 persuasion scenarios. Note that these categories overlap. We will include this distribution in the paper.
>
> *"The framework and the evaluation mainly focus on two agents. It is not trivial to extend the interaction and evaluation to multi-agent coordination. How can we scale the interaction and evaluation to more complex social scenarios?"*
>
> We agree with the reviewer that extending Sotopia to multi-agent (>2 agent) scenarios is an interesting topic that future work should explore. From a practical perspective, SOTOPIA, built on the PettingZoo framework, could support interactions among multiple agents. The only additional feature required would be managing turn-taking, which could be done via a number of methods, such as bidding [1]. Evaluating multi-dimensional social intelligence for each character in multi-agent scenarios can be trivially conducted in SOTOPIA.
>
> [1] Selfridge, Ethan O., and Peter A. Heeman. 2010. “Importance-Driven Turn-Bidding for Spoken Dialogue Systems.” In Proceedings of the 48th Annual Meeting of the Association for Computational Linguistics, 177–85. ACL ’10. USA: Association for Computational Linguistics.

---

> ### Author Response · Authors · 2023-11-22
> **Further explanations of biases in evaluating with GPT-4**
>
> Thank you for further illustrating the important issue of potential bias in GPT-4 for evaluation. We acknowledge this concern and have conducted various analyses, as well as referenced various previous works, to highlight the possible risks. Specifically, Figure 2 shows the distributional differences between GPT-4 scores and human scores, which indicates *“GPT-4 is more likely to rate higher instead of lower than humans when it disagrees with average human judgment”*.
>
> Expanding on that, we have a more fine-grained breakdown analysis by each social dimension in Appendix Figure F.1/2/3, which further shows the effect of GPT-4 being more lenient than human annotators on machine-generated interactions. Furthermore, we have conducted an in-depth analysis of the biases in GPT-4's evaluations across different models. As shown in the table, interestingly, GPT-4 is actually more lenient to social interactions generated by LLaMa2 and GPT-3.5 compared to those generated by itself.
>
> As you mentioned, this does not hurt using GPT-4 to *“show the relative strength of each dimension of an agent”* but rather highlights a limitation in using *“be an optimization target”* (which is out of the scope of this paper). We would like to further illustrate the importance of human annotation and as an initial step, one can calibrate the GPT-4 evaluation by subtracting the mean differences outlined below. Nevertheless, it's important to note that this remains a complex challenge [1]. We encourage future research to explore the potential of using GPT-4's scores as rewards in further studies.
>
> |       | BEL  | REL | KNO  | SEC  | SOC  | FIN  | GOAL |
> |-------|------|-----|------|------|------|------|------|
> | GPT-4 | 1.44** | 0.13 | 0.56* | 0.17* | 0.32** | 0.13 | 0.88** |
> | GPT-3.5 | 2.18** | 0.21* | 1.06** | 0.11 | 0.50** | 0.25** | 1.48** |
> | Llama2 | 3.43** | 0.05 | 0.91** | 0.24 | 0.56** | 0.14 | 1.13** |
>
> Mean = avg_GPT-4_evaluation-avg_human_evaluation
> \** means a 0.01 significance level under t-test
> \* mean a 0.05 significance level under t-test
>
> [1] Pernot, Pascal and Fabien Cailliez. “A critical review of statistical calibration/prediction models handling data inconsistency and model inadequacy.” Aiche Journal 63 (2016): 4642-4665.

---

> > ### Comment · Reviewer_qjhX · 2023-12-04
> > **Thanks for the explanation of GPT-4 biases!**
> >
> > I would like to thank the authors for discussing the differences between GPT-4 ratings and human ratings, and talking about ideas for using human ratings as calibration. I agree that handling the inconsistency between human ratings and model ratings is a complex issue. But as a new evaluation, I would suggest the authors explicitly discuss how the community should use and treat the ratings from the GPT-4 evaluations when running SOTOPIA. I'll update my review to acknowledge the authors' explanation.

---

### Public Comment · ~Guohao_Li1 · 2023-11-14
**Missing related work**

The paper introduces "SOTOPIA," an environment for simulating complex social interactions between artificial agents and evaluating their social intelligence. It involves agents (both artificial and human) role-playing various characters in different social scenarios to achieve complex goals including cooperative, competitive, and mixed social goals. The evaluation framework, "SOTOPIA-EVAL," assesses performance across multiple dimensions, including goal completion, believability, and knowledge acquisition.

Thanks for the great work.  However, it could also be beneficial to discuss prior work on multi-LLM agents and role-playing for the study of cooperative settings [1].

[1] Li, Guohao, Hasan Abed Al Kader Hammoud, Hani Itani, Dmitrii Khizbullin, and Bernard Ghanem. "CAMEL: Communicative Agents for" Mind" Exploration of Large Language Model Society." NeurIPS 2023

---

> ### Author Response · Authors · 2023-11-15
>
> Hi Guohao,
>
> Thanks for your comment. CAMEL is super relevant, and I think we planned to discuss it, but it was accidentally removed during editing. We  will discuss CAMEL in the revision.

---

### Author Response · Authors · 2023-11-22
**Revision**

We have revised our submission based on reviewer's comments:

1. Added a few missing citations
2. Fixed various typos, grammar errors and explicitly stated the test used in Table 2.

We will update our submission again after we get all the empirical results mentioned in the rebuttal and discussion.

Thank you for your comments and suggestions!

---

### Meta-Review · Area_Chair_3Rm2 · 2023-12-06

**Metareview:**

The reviewers generally appreciate the work, commenting on the interactivity of the proposed framework (vs traditional static evaluations), clarity of presentation, and timeliness.

They do raise several concerns, such as bias introduced by using GPT-4 as part of the evaluation, comparison to related efforts, and details regarding the metrics.

The authors provide comprehensive responses and run additional experiments when needed, which seem to address the comments.

**Justification For Why Not Higher Score:**

Although it is a nice paper, there are some points for improvement (e.g. reliance on GPT-4).

**Justification For Why Not Lower Score:**

This is a comprehensive benchmark for social interactions, different from related works that statically evaluate dialogues post-hoc.

---

### Decision · Program_Chairs · 2024-01-16

Accept (spotlight)